# Analytical and computational workflow for in-depth analysis of oxidized complex lipids in blood plasma

Angela Criscuolo[2,3,4,9], Palina Nepachalovich[1,2,3,9], Diego Fernando Garcia-del Rio[2,3], Mike Lange[2,3], Zhixu Ni [1,2,3], Massimo Baroni [5], Gabriele Cruciani [6], Laura Goracci[6], Matthias Blüher [7,8] & Maria Fedorova [1,2,3] ✉

Lipids are a structurally diverse class of biomolecules which can undergo a variety of chemical modifications. Among them, lipid (per)oxidation attracts most of the attention due to its significance in the regulation of inflammation, cell proliferation and death programs. Despite their apparent regulatory significance, the molecular repertoire of oxidized lipids remains largely elusive as accurate annotation of lipid modifications is complicated by their low abundance and often unknown, biological context-dependent structural diversity. Here, we provide a workflow based on the combination of bioinformatics and LC-MS/MS technologies to support identification and relative quantification of oxidized complex lipids in a modification type- and position-specific manner. The developed methodology is used to identify epilipidomics signatures of lean and obese individuals with and without type 2 diabetes. The characteristic signature of lipid modifications in lean individuals, dominated by the presence of modified octadecanoid acyl chains in phospho- and neutral lipids, is drastically shifted towards lipid peroxidation-driven accumulation of oxidized eicosanoids, suggesting significant alteration of endocrine signalling by oxidized lipids in metabolic disorders.

Lipids are an extremely diverse class of small biomolecules, for which the whole natural variety is yet not fully defined. So far more than 47,000 of lipid molecular species are annotated (LIPID MAPS; June 2022), and over hundreds of thousands being computationally predicted. In addition to the high diversity of original species derived from classical biosynthesis routes, lipids can undergo various modifications by the introduction of small chemical groups via enzymatic and non-enzymatic reactions, forming higher levels of structural complexity, termed epilipidome[1]. Modifications of lipids including oxidation, nitration, and halogenation are described for numerous physiological and pathological conditions and are generally linked to the regulation of various signaling events[2]. Thus, epilipidome, in a manner similar to epigenetic and post-translational protein modifications, increases the regulatory capacity of biological systems by supporting prompt responses to different stimuli and stressors.

Among different modifications, lipid (per)oxidation was so far studied the most. Oxylipins, oxygenated derivatives of polyunsaturated fatty acids (PUFAs), are commonly analyzed as markers of systemic and tissue-specific inflammation and its resolution[3]. Although usually measured in the form of free fatty acids, the majority of oxylipins in vivo are believed to be esterified to complex lipids, including glycerophospholipids (GPLs), cholesteryl esters (CEs), and triglycerides (TGs)[4]. A positive correlation between endogenous levels of oxidized complex lipids and the development and progression of numerous human diseases including cardiovascular (CVD), pulmonary and neurological disorders as well as non-alcoholic steatohepatitis (NASH) was

demonstrated[5–8]. Oxidized lipids have been shown to accumulate in apoptotic cells as well as microsomes released by activated or dying cells, and are generally associated with inflammatory responses, mediated mainly via the innate immune system[2]. Inflammation modulating potential of oxidized lipids is attributed to their recognition by and thus activation of a wide range of pattern recognition receptors including macrophage scavenger receptors, Toll-like receptors, CD36, and C reactive protein[9]. Moreover, oxidized lipids and their protein adducts can be recognized by natural IgM antibodies and thus sequestered from the circulation[5]. Furthermore, the recently formulated concept of ferroptosis placed oxidized membrane GPLs in the center of the cell death execution program via necrotic rupture of cellular plasma membrane[10].

Despite overwhelming amounts of data on their biological significance, the pathways leading to the intracellular and extracellular generation of oxidized complex lipids and thus their in vivo structural diversity remain largely elusive. Both enzymatic (via action of lipoxygenase, and cytochrome P450 systems[11,12]) and non-enzymatic (via free radical-driven lipid peroxidation (LPO)[13,14]) systems are proposed to contribute to the accumulation of oxidized lipids. Overall, an increasing body of evidence proposes that action of oxidized complex lipids is context-dependent and their bioactivities might be manifested either in an adaptive or a pathological manner[9,11]. Understanding the underlying regularities in epilipidomics patterns would require holistic mapping of oxidized lipids in different conditions in a variety of biological matrices. Liquid chromatography coupled on-line to mass spectrometry (LC-MS) was actively applied for the identification and quantification of oxidized complex lipids with a main focus on oxidized phosphatidylcholines (oxPCs) and oxCEs, with several reports also addressing modifications of phosphatidylethanolamines (oxPEs)[15,16], phosphatidylserines (oxPSs)[17] and oxTGs[18]. With just a few reports addressing the high diversity of oxidized GPLs[15,19,20], the vast majority of the available LC-MS-derived data report only a handful of oxidized species analyzed by targeted LC-MS workflows. Indeed, due to their low natural abundances, analysis of oxidized lipids in complex biological matrices mostly relies on targeted LC-MS utilizing a limited list of previously identified oxidized species independent of the biological context preventing the discovery of sample-specific epilipidome.

Here we describe the LC-MS/MS workflow developed for comprehensive, biological matrix-specific epilipidome profiling and relative quantification. The method combines biological intelligence-driven in silico prediction of a sample-specific epilipidome followed by its semi-targeted detection using a set of optimized LC, MS, and MS/MS parameters aiming to increase the accuracy of the annotation. The developed workflow, validated using blood plasma samples of lean, non-diabetic and type 2 diabetic obese individuals, allowed the identification and relative quantification of oxidized PC, CE, and TG lipids, showing both physiological (lean) and pathological (obese) patterns, thus supporting endocrine signaling role of the oxidized epilipidome.

## Results and discussion

### Fragmentation rules for annotation of oxidized complex lipids

Mass spectrometry-based annotation of oxidized lipids requires accurate detection of the precursor ion $m/z$ (MS1) from which lipid elemental composition can be deduced, as well as an informative fragment mass spectrum (MS2) allowing the assignment of lipid class, molecular species, modification type, and modification position. For instance, oxGPLs ionized in negative ion mode upon collision-induced dissociation (CID) produce intense fragment ions specific to the head groups, and fatty acyl chains, including the one carrying the modification. Studies on oxygenated free fatty acids (oxylipins) demonstrated that the anion of oxygenated fatty acid upon CID undergoes charge-remote fragmentation resulting in a set of fragments characteristic to the modification type and position along the hydrocarbon chain[21–24]. Recently, to induce similar fragmentation patterns for oxidized complex GPLs, multistage fragmentation techniques (MS3) on tribrid MS

instruments were applied for the annotation of oxidized PC and PE lipids in complex biological samples[25] (Fig. 1a). However, the multistage fragmentation reduces sensitivity (when MS3 spectra are recorded in the orbitrap) and/or resolution (when the spectra are recorded in the ion trap). Here, using a set of in vitro oxidized PC lipid standards, we demonstrate that MS2 spectra obtained using elevated energy HCD display a similar set of fragment ions without the need of multistage ion activation (Fig. 1b). Thus, HCD of oxPC formate adduct ion at $m/z$ 818.5545 obtained using stepped normalized collision energy of 20-30-40 units, provided lipid class (head group-specific ions at $m/z$ 758.5352, 168.0431 and 224.0695), molecular species (anions of fatty acyl chains at $m/z$ 255.2332 and 295.2281), modification type (water loss characteristic to the presence of hydroxyl group at $m/z$ 277.2175), and modification position ($m/z$ 195.1392, the product of cleavage of the adjacent to the carbinol C-C bond) specific fragment ions allowing annotation of this oxidized lipid as PC(16:0_18:2<OH{13}>). Similarly, each type of modification and its positional isomers in oxGPLs can be characterized by a set of specific fragment ions obtained either in MS3 or elevated energy HCD MS2 experiments.

Unfortunately, chemically defined standards of oxidized complex lipids necessary to create a fragment spectra library, are limited to just a few commercially available molecular species and do not reflect the diversity of possible endogenous analytes. To facilitate further high-throughput annotation of oxGPLs, we compiled our results obtained by MS3 and elevated energy stepped HCD MS2 fragmentation of selected oxylipin standards and in vitro oxidized PC lipids (Supplementary Data 1.1 and 1.2), literature data on the fragmentation of oxidized free fatty acids and complex lipids[20,22,26–34], as well as available MS2 spectra from METLIN, LIPID MAPS, and MS DIAL.msp library, in the form of fragmentation rules exemplified here for different modification types and positions on oxidized oleoyl (18:1), linoleoyl (18:2), and arachidonoyl (20:4) chains in PC lipids (Supplementary Data 2).

To extend these rules beyond oxGPL lipids, we performed in vitro oxidation of neutral lipids, namely CE and TG. Neutral lipids do not ionize in negative ion mode and are usually monitored as positively charged ammonium adducts. However, when in vitro prepared standards were analyzed by LC-MS/MS, we noticed that upon oxidation both CE and TG have a higher tendency to form sodiated, and in some cases protonated, adducts (Fig. 1c). Moreover, adduct preferences were modification type-specific (Fig. 1d, e). Thus, unmodified CEs were preferentially ionized as $NH_4$ adducts (83% of total abundance) with minor contribution of sodiated forms (15%), whereas CE hydroperoxides showed preference towards Na adduct formation (61%). The preference for Na adduct was even more evident for hydroxylated CE derivatives with 87% detected in the form of sodiated ions. The majority of epoxyCE, however, remained in ammoniated forms (55%), and oxoCE derivatives equally formed sodiated and protonated (48%) adducts (Fig. 1d). Oxidized TG displayed overall similar modification type-specific adducts distribution; however, ammoniated adducts remained dominant for almost all modification types (Fig. 1e).

Considering the trend of oxCE and oxTG to form differential adducts upon electrospray ionization (ESI), we further compared the MS2 spectra of in vitro oxidized standards obtained from different precursor ions in terms of their utility towards accurate oxidized lipids annotations. Protonated species provided non-informative MS2 (data not shown), whereas both Na and $NH_4$ adducts showed some structure-related fragments (Fig. 2a). Thus, the HCD spectrum of CE(18:2<OOH>) ammonium adduct displayed a single meaningful fragment ion corresponding to the protonated form of cholestadiene, whereas the sodiated precursor resulted in much more informative MS2 spectrum with modification type- and position-specific neutral loss and fragment ions. Indeed, alkali metal adducts of oxidized lipids and oxylipins were previously reported to follow the similar charge-remote fragmentation mechanism described above for oxidized fatty acyl chain anions[21–23]. Here, we further optimized HCD collision energy

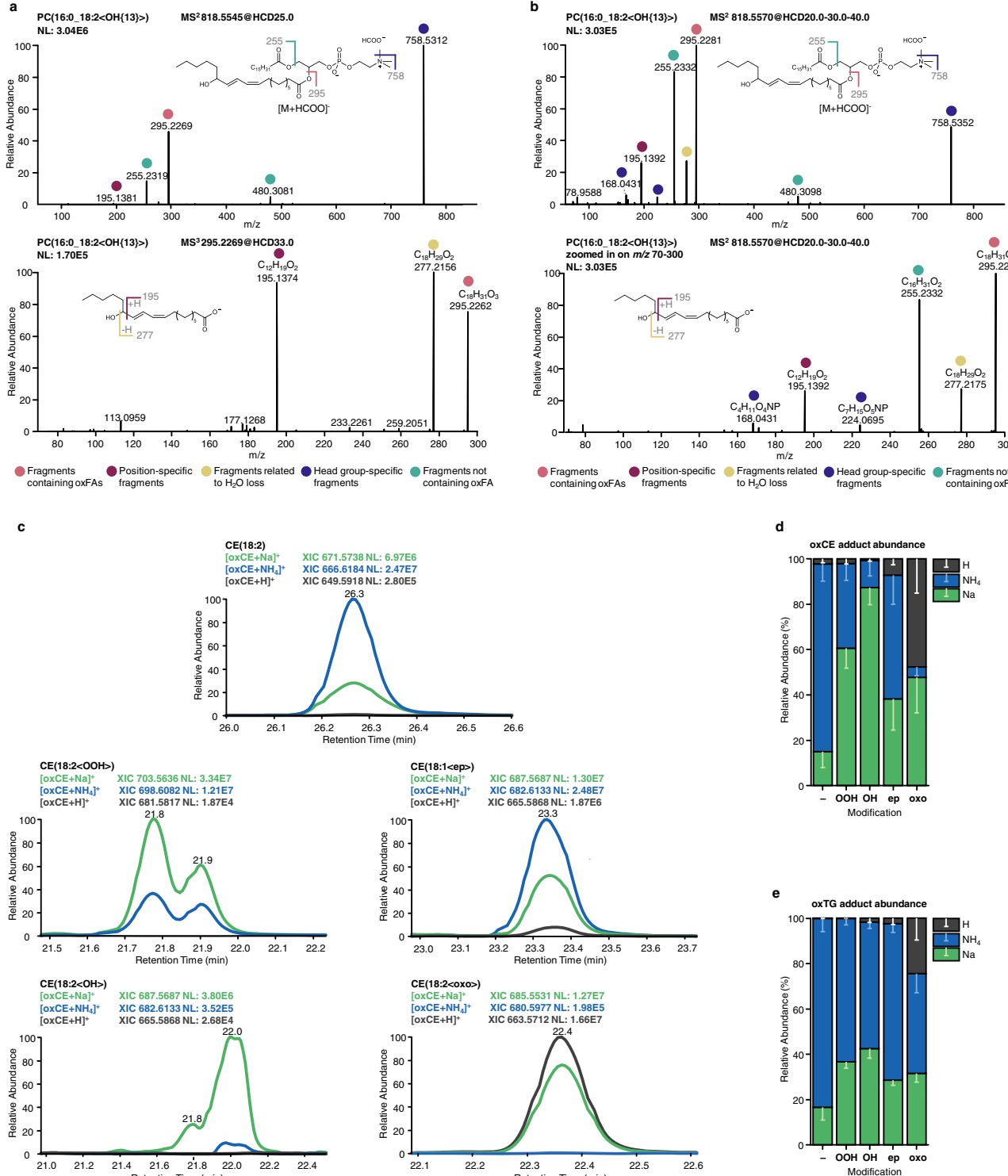

**Fig. 1 | Optimization of MS and MS2 parameters for the structural elucidation of oxidized complex lipids structures.** Tandem mass spectrometry analysis of oxidized PC lipid using multistage MS3 (**a**) or optimized elevated energy stepped HCD MS2 experiments (**b**) both allowed structural assignment of PC(16:0_18:2<OH{13}>). Positive ion mode ionization adducts for in vitro oxidized neutral lipids showed modification type-specific preferences. Qualitative (**c**; illustrated using extracted ion chromatograms for oxCE(18:2) species) and quantitative (**d**, **e**) distributions illustrate modification-specific preferences to form protonated (dark gray), ammoniated (blue) or sodiated (green) adducts for unmodified (-), hydroperoxy- (OOH), hydroxy (OH), epoxy (ep), or keto (oxo) derivatives of CE and TG. Relative abundances of differential adducts for oxCE (**d**) and oxTG (**e**) represent (mean ± SD) values calculated for 10 species. Source data are provided as a Source Data file.

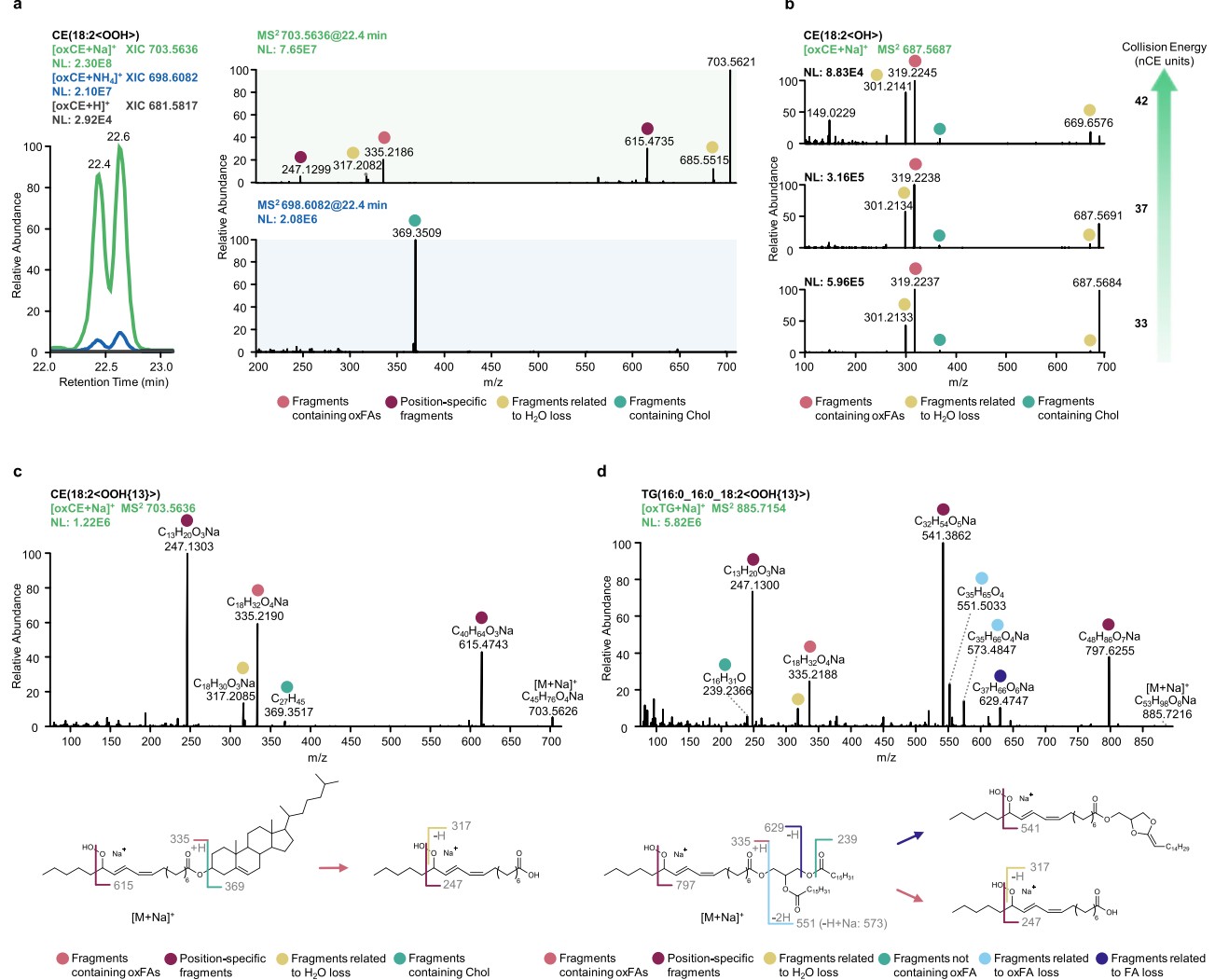

**Fig. 2 | Optimization of elevated energy HCD of sodiated precursors to support structural elucidation of oxidized neutral lipids.** Utility of tandem mass spectra to provide structure-relevant fragment ions was compared for protonated (gray), ammoniated (blue) and sodiated (green) oxCE precursor ions (**a**). Considering larger numbers of structure-relevant fragment ions obtained by HCD of sodiated oxCE precursors, the collision energy was optimized to enhance fragment ions formation (**b**). Elevated energy stepped HCD MS2 experiments using sodiated precursors were shown to provide highly informative spectra, exemplified here for CE(18:2<OOH{13}>) (**c**) and TG(16:0_16:0_18:2<OOH{13}>) (**d**).

to obtain informative MS2 patterns allowing the identification of both modification type- and position-specific fragment ions of sodiated oxCE and oxTG. Overall, Na adducts were more stable than their ammoniated counterparts and required higher collision energies for their efficient fragmentation (Fig. 2b). Elevated energy stepped HCD of 30-40-50 normalized units was chosen for further experiments and allowed to define fragmentation rules for hydroperoxy derivatives of CE and TG lipids (Fig. 2c, d) as well as other types of modifications (Supplementary Data 1.3 and 1.4). Similar to oxPC, we compiled MS2 data on sodiated precursors from in vitro oxidized CE/TG as well as available literature data in the form of modification and position-specific fragmentation rules to be used for high-throughput annotation of modified neutral lipids (Supplementary Data 2).

To validate ionization preferences and fragmentation rules in complex biological matrices, we performed in vitro oxidation of commercially available human blood plasma and were able to confirm the presence of the described adduct types and to annotate selected oxidized species using a set of fragmentation rules provided in Supplementary Data 2 (Supplementary Data 3). Taken together, using in vitro oxidized lipid standards and human blood plasma, we defined the preferential ionization adducts to be utilized for MS2 experiments

aiming for the identification of oxidized complex lipids. The optimized elevated energy HCD was shown to be efficient in providing informative MS2 spectra of oxidized lipids. Charge-remote fragmentation of acyl anions or their sodiated analogues in oxPC and oxCE/oxTG, respectively, allowed efficient assignment of hydroxy-, epoxy-, keto-, and hydroperoxy-modified lipids as well as oxidatively truncated forms in a modification type and position-specific manner. The fragmentation rules defined for each modification type and position can be further used to support high-throughput annotation of lipid modifications.

### Increasing annotation accuracy by retention time mapping

Lipid oxidation might lead to the formation of a large number of isomeric species including isomers carrying different types of modifications as well as positional isomers. Although unresolvable by MS, isomeric oxidized lipids can be separated by reversed-phase chromatography (RPC)[35]. Here, to characterize in vitro oxidized standards and blood plasma we applied C18 RPC coupled on-line to MS. LC-MS coupling allowed separation of multiple isomeric oxidized lipids providing MS2 spectra sufficient for accurate the annotation. Moreover, we were able to define the elution order for mono- and dioxygenated isomeric species (Fig. 3a). The elution order was consistent within

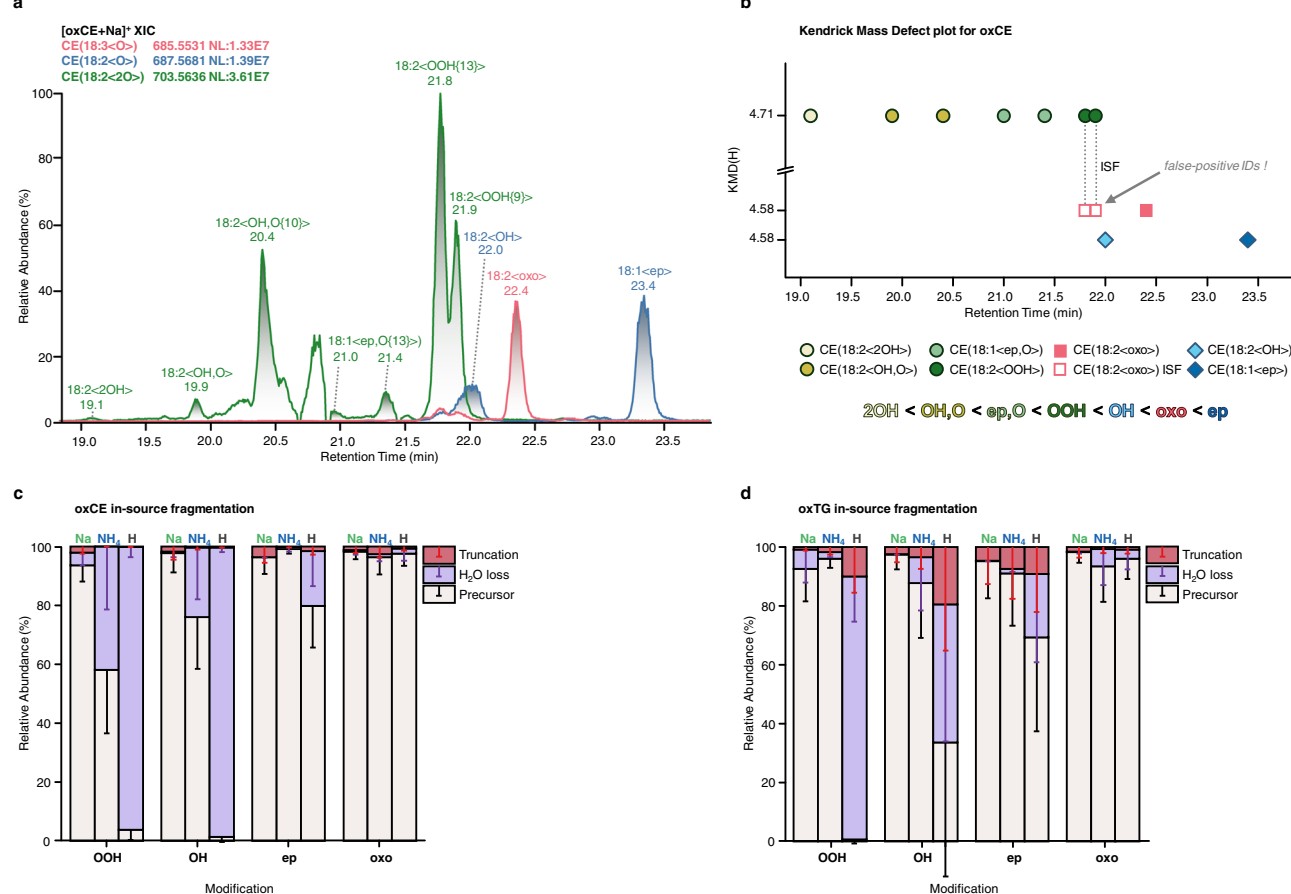

**Fig. 3 | LC-MS/MS coupling and retention time (RT) mapping facilitate resolution of isomeric oxidized lipids and allow to correct for in-source fragmentation (ISF).** Isomeric oxidized lipids can be successfully resolved by RPC, as illustrated here for mono- (CE(18:3<O>) in red and CE(18:2<O>) in blue) and dioxygenated (CE(18:2<2O>) in green) oxCE lipids (**a**). RT mapping using KMD(H) plots allows visual inspection of isomer-specific elution order as well as identification of false positive annotation due to the ISF (**b**). The extent of ISF for oxCE (**c**) and oxTG

(**d**) lipids was shown to be ionization- and modification type-specific. ISF was calculated based on the parent and product peak areas using the formula

$$ISF = \frac{\text{Peak Area}_{Product}}{(\text{Peak Area}_{Product} + \text{Peak Area}_{Parent})} \times 100.$$ Relative abundances of ISF-derived species (truncation and water loss) for oxCE (**c**) and oxTG (**d**) represent (mean ± SD) values calculated for 10 species. Source data are provided as a Source Data file.

different lipid classes (PC, CE, TG, and oxylipin standards) and was identical to the one described for isomeric oxylipins in the literature[30,36]. Thus, for RPC-separated monooxygenated lipids the elution order was OH < oxo < epoxy, and for dioxygenated it was 2OH < OH,O < ep,O < OOH. Dioxygenated lipids eluted earlier than their monooxygenated counterparts. For regioisomers the general trend was defined as well, with earlier elution times for the species carrying the modification closer to the ω-end of the acyl chains (e.g. Fig. 3a for C13-OOH < C9-OOH, and Supplementary Fig. 1 for monooxygenated PC(16:0_20:4<OH>) isomers). Baseline separation of regioisomers was not possible for majority of the species, however use of extracted chromatograms of position-specific fragment ions allowed accurate assignment of corresponding retention times (RTs). Furthermore, previously introduced RT mapping of oxidized lipids using Kendrick mass defect (KMD) plots[37] was employed here for visualization and manual inspection of the defined elution order to increase accuracy of the annotations in high-throughput epilipidome profiling experiments (Fig. 3b).

Importantly, using KMD vs RT plots we were able to spot previously unconsidered in-source fragmentation (ISF) of oxidized lipids. Despite the application of soft ionization techniques such as ESI, lipid ions remain quite labile and can undergo undesired fragmentation via collision with neutral gas molecules in the intermediate region of MS instruments between the atmospheric pressure ion source and deep

vacuum environment of the mass analyzer. Those fragment ions are often identical to the endogenous lipids of other subclasses and thus ISF is a known source of false positive identifications in conventional lipidomics[38,39]. Similarly, we demonstrated that oxidized lipids can undergo substantial ISF with the formation of ions related to other types of modifications. For instance, CE hydroperoxides undergo ISF accompanied by the loss of water with the formation of the ion with m/z equal to the endogenous oxoCE derivative (Fig. 3a, red trace; and Supplementary Fig. 2). As ISF occurs after LC separation, it can be recognized and thus corrected by RT mapping in the form of KMD vs RT plots (Fig. 3b, open red squares).

Using this approach, we systematically evaluated the extent of ISF for different type of adducts and modifications for oxPC, oxCE, and oxTG. At the settings for the ion transfer optics used in these experiments, oxPC did not display any significant ISF, whereas both oxCE and oxTG showed adduct- and modification type-specific ISF. Among the most significant fragmentation events driven by the presence of the modification were the loss of water and gas-phase acyl chain cleavage (truncation) at the site adjacent to the modification. Protonated precursors underwent the most significant ISF for all studied modifications except oxo-derivatives (Fig. 3c, d), which probably explains the observed adduct abundance described above for this modification type (Fig. 1c, d). Similarly, in line with the observation that sodiated adducts required higher collision energies to obtain fragment-rich

MS2, Na adducts of both oxCE and oxTG were more resistant to ISF (Fig. 3c, d, Supplementary Fig. 2), whereas ammoniated adducts showed ISF up to 42%. Overall sodiated adducts showed highest analytical robustness and thus selectivity of oxidized lipid profiling, although they are not necessarily represented as the most intense species. Among the modification types, hydroxy- and hydroperoxy-modified CE and TG were the most labile whereas epoxy- and keto-derivatives generally showed high stability. Thus, we demonstrated the efficiency of RT mapping to increase the accuracy of oxidized lipid annotations both in terms of isomer resolution and the correction for ISF. Taken together, we provided multi-level annotation guidelines for oxPC, oxCE, and oxTG lipids based on the selection of the preferential ionization adducts showing minimized ISF (MS1 level) and the most informative MS2 patterns, modification type- and position-specific fragmentation rules (MS2 level), as well as RT mapping (LC level) to be used for the accurate and high-throughput identification of oxidized complex lipids from LC-MS/MS datasets.

## Semi-targeted LC-MS/MS for sample specific epilipidome profiling

Multi-level (LC-MS1-MS2) annotation guidelines developed above could support identification of oxidized complex lipids in biological matrices but only when high quality MS2 spectra are made available. Application of targeted MS methods limits epilipidome coverage to a predefined subset of oxidized species which do not necessarily reflect the biological context of the sample in question. On the other hand, the application of untargeted data-dependent (DDA) method showed to be generally inefficient for the detection of oxidized lipids as their low endogenous abundance prevents them to be selected for MS2 events in the presence of highly abundant unmodified lipids. To find a compromise between targeted solution limiting epilipidome coverage and undersampling of low abundant oxidized lipids by untargeted DDA, we developed a semi-targeted LC-MS/MS method. The method relies on the combination of in silico prediction of a sample specific (e.g. human blood plasma, liver, or particular cell type) epilipidome which is used as an inclusion list for semi-targeted DDA (stDDA) considering MS1 (preferential adduct selection) and MS2 (lipid class specific elevated energy HCD) settings optimized as described above. For the method development and validation, we used lipid extracts from blood plasma of lean non-diabetic (LND), obese non-diabetic (OND) and obese with type 2 diabetes (OT2D) individuals for whom endogenous oxidized complex lipids with largely unknown diversity are expected.

Considering that the native lipidome serves as the substrate for lipid (per)oxidation, epilipidome of a particular tissue can be predicted from its native lipidome. In blood, lipids are transported mainly in the form of lipoproteins with PC, CE, and TG representing the main lipid classes. We and others previously reported the lipid composition of human blood plasma[40,41], from which here we selected 45 most abundant PUFA-containing PC, CE, and TG lipid molecular species to be used as sample-specific endogenous substrates for in silico oxidation (Fig. 4, Step 1). A new version of LPPtiger software (LPPtiger 2.0) was employed to predict the human blood plasma-specific epilipidome using 17 PC, 17 TG, and 11 CE lipid molecular species as the substrates (Supplementary Data 4). Rather than performing simple enumeration with oxygen atoms, the in silico oxidation algorithm within LPPtiger 2.0 relies on molecular networks manually reconstructed for 10 PUFAs based on the literature data available on their enzymatic and non-enzymatic oxidation products[42]. Moreover, a user can specify the level of in silico oxidation which was here constrained to the maximum of two oxidation sites with up to 2 hydroxy, 2 epoxy, 1 oxo, or 1 hydroperoxy modifications in long-chain and truncated oxidized lipids, thus limiting the predicted chemical space to the most probable lipid modifications. Using 45 native lipids as in silico oxidation substrates, LPPtiger 2.0 predicted 956 oxidized species corresponding to 559 unique elemental compositions (Fig. 4, Step 2, Fig. 5a). Those were

used to compose inclusion lists for stDDA LC-MS/MS analysis considering their preferential ionization mode and adducts defined above for each lipids class (Fig. 4, Step 3). Thus, oxPCs were analyzed in negative ion mode as formate adducts, and oxCEs/oxTGs as sodium adducts in positive ion mode using elevated energy HCD levels specific for each lipid class. Considering efficient RPC separation of oxidized lipid classes, we further used polarity switching with the 1st half of the LC gradient measured in negative (oxPC) and the 2nd in positive (oxCE and oxTG) ion modes to facilitate higher throughput of the analysis. Using this setup 559 unique m/z for predicted oxidized lipids were targeted within two LC-MS/MS DDA analysis per group-specific plasma pool (e.g. LND, OND, and OT2D) using a sample amount equivalent to as low as 1.3 μL of blood plasma per sample (Fig. 4, Step 3).

The developed semi-targeted method provided several important advantages including large coverage of the epilipidome predicted in a sample-specific manner (Fig. 5a). The use of stDDA instead of multiple reaction monitoring (MRM) or parallel reaction monitoring (PRM) for initial identification of oxidized lipids allowed profiling of a larger number of analytes without repetitive sample injections. Indeed, within the large inclusion list of possible targets, only those above certain intensity threshold and abundance rank (here top 6 mode was used) were selected for MS2 ensuring sufficient quality of MS2 spectra, without wasting instrument cycle time on extremely low intense or even absent analytes, fragmentation of which would not result in senseful MS2 spectra anyway. The use of a relatively low number of precursors for MS2 selection (top 6) within each DDA cycle further allowed to increase injection time (200 ms) to accumulate larger number of ions for the fragmentation without compromising the resolution provided by RPC. Elevated energy HCD provided informative MS2 spectra without the need of multistage ion activation (MS3) thus increasing the sensitivity and resolution necessary for the accurate annotation of isobaric modification type- and position-specific fragments. Moreover, RT-scheduled polarity switching allowed to record MS2 spectra for both oxPC and oxCE/oxTG within the same LC-MS/MS analysis. Taken together, the developed stDDA method allowed profiling of oxidized complex lipids in a comprehensive, biological matrix-specific, and sensitive way.

## Oxidized complex lipids in blood plasma of lean and obese individuals

Using the lipidome-specific prediction and stDDA method described above, we analyzed group-specific pooled samples (Fig. 4, Step 3) in which 339 oxidized complex lipids corresponding to 78 unique m/z were identified (Fig. 5a; Supplementary Data 5-8). The initial identification of modified lipids was performed by LPPtiger 2.0, followed by the manual annotation based on the fragmentation rules to define modification type and position-specific species (Supplementary Data 2). A large number of isomeric oxidized lipids were annotated and confirmed using elution rules and RT mapping via KMD(H) plots (an example for oxCE is presented in Supplementary Fig. 3).

Having identified oxidized complex lipids in group-specific pooled samples, we further performed their relative quantification in individual blood plasma lipid extracts of LND (n = 50), OND (n = 50) and OT2D (n = 50) individuals. To this end, the targeted PRM method was employed utilizing RT scheduling and polarity switching (Fig. 4, Step 4). Out of 339 identified oxidized complex lipids, 116 species (46 unique m/z) were quantified (Fig. 5a; Supplementary Data 9).

Out of 116 quantified oxidized lipids, 20 oxPC, 26 oxTG and 26 oxCE species showed significant differences between three groups of samples (ANOVA p < 0.05) (Fig. 5 b–d, Supplementary Data 10). Interestingly, out of these 72 oxidized lipids, 25 showed higher levels in LND relative to obese (OND and OT2D) individuals. Importantly, those lipids were specifically represented by octadecanoids, a family of oxidized lipids bearing 18 carbon-long fatty acyl chains. Thus, oxidized complex lipids enriched in the blood plasma of LND individuals exclusively

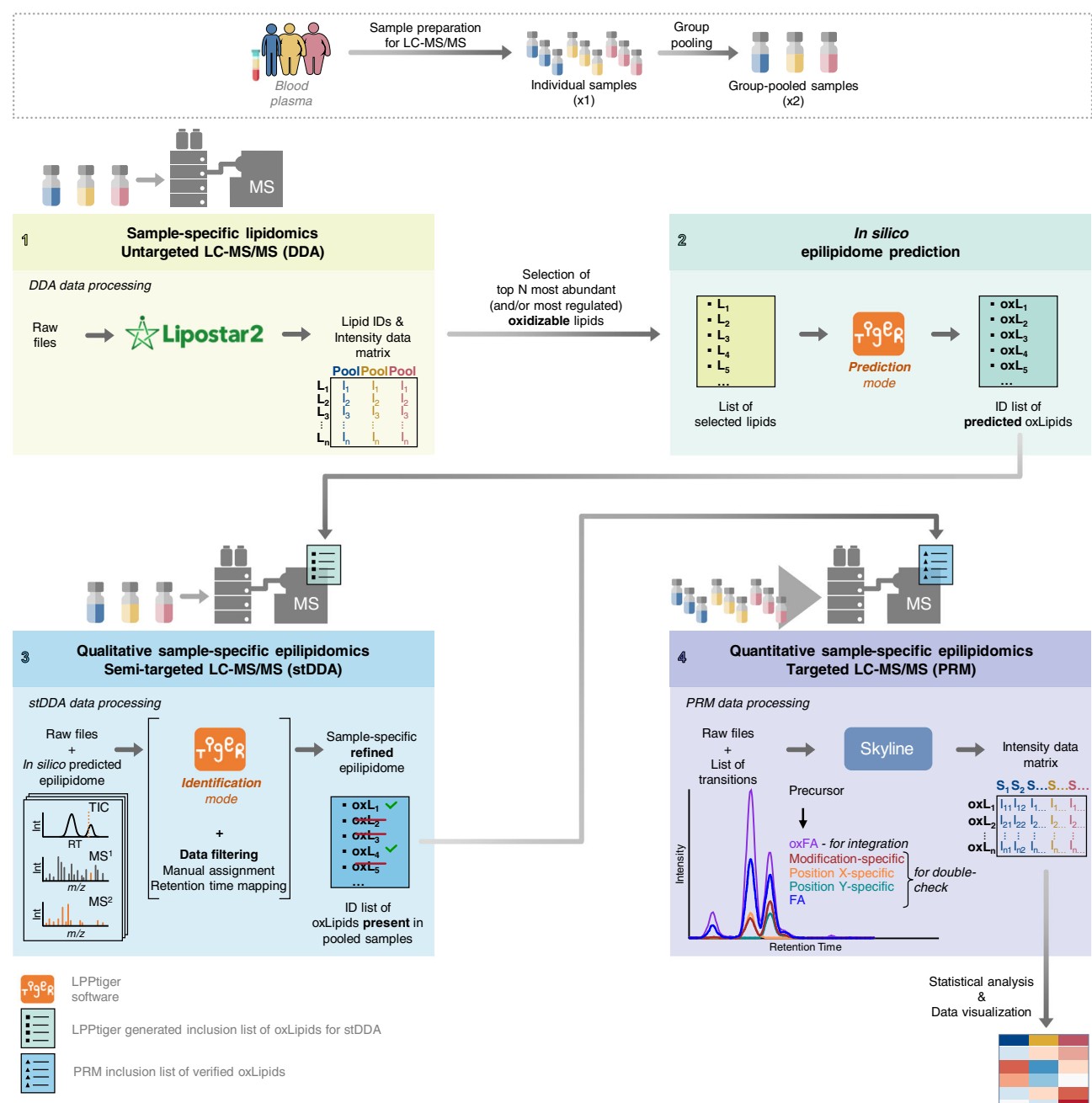

**Fig. 4 | Schematic representation of analytical and bioinformatics workflow developed for identification and relative quantification of oxidized complex lipids.** Blood plasma lipids are extracted and experimental group-specific pooled samples are prepared by mixing equal amounts of each individual sample within the group. Step 1: Group-specific pooled samples are used for untargeted LC-MS/MS analysis (e.g. using data-dependent acquisition, DDA) to identify native, non-oxidized lipidome of biological system in question (here, human blood plasma). Data processing for identification of native lipidome can be done by any suitable software of user choice (e.g. Lipostar 2). Step 2: Most abundant and/or most regulated lipids containing polyunsaturated acyl chains (and thus susceptible to oxidation) are selected for in silico epilipidome prediction. In silico prediction step is fully automated within LPPtiger 2.0 software (for details see methods section as well as User Manual available at https://github.com/LMAI-TUD/lpptiger2). Step 3: LPPtiger 2.0-generated inclusion list containing *m/z* values of predicted epilipidome can be directly imported in a semi-targeted DDA (stDDA) method template and used to analyze group-specific pooled samples. stDDA raw files are then analyzed by LPPtiger 2.0 to identify oxidized complex lipids. Manual inspection of MS2 spectra and RT mapping (e.g. by using KMD plots) are advised to support accurate annotation of modification type- and position specific-isomers. Step 4: List of identified species can be exported by LPPtiger 2.0 in a form of PRM/MRM lists for targeted analysis of oxidized lipids in individual samples. Targeted dataset can be processed by any dedicated software (e.g. here Skyline). Obtained results can be worked up by any available statistical/visualization tools.

carried modified 18:1, 18:2 and 18:3 acyl chains. Out of 10 oxPC species enriched in LND vs obese individuals, 9 corresponded to the mono-oxygenated octadecanoids (Fig. 5b). Unfortunately, due to the extremely low abundance of those lipids, the confident structural assignment of the modification type and position was not always possible. However, we were able to perform relative quantification for PC(16:0_18:2<OH>) and PC(18:0_18:2<OH>) regioisomers (Supplementary Fig. 4a, b). 13-Hydroxyoctadecanoid acid (HODE)-containing isomers were significantly higher than 9-HODE in all three groups of samples (LND, OND, and OT2D) pointing to the role of

**a**

| Epilipidome (ID / unique m/z) | oxPC | oxCE | oxTG | Total |
|---|---|---|---|---|
| Unmodified lipid precursors | 17 | 11 | 17 | 45 |
| *In silico* predicted | 447 / 210 | 217 / 162 | 393 / 187 | 956 / 559 |
| Identified | 75 / 21 | 136 / 34 | 121 / 16 | 339 / 78 |
| Quantified | 40 / 13 | 40 / 19 | 36 / 14 | 116 / 46 |
| Significant (ANOVA p < 0.05) | 20 / 10 | 26 / 14 | 26 / 12 | 72 / 36 |

**b**

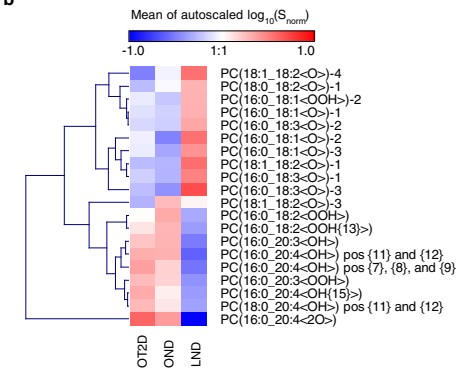

**c** 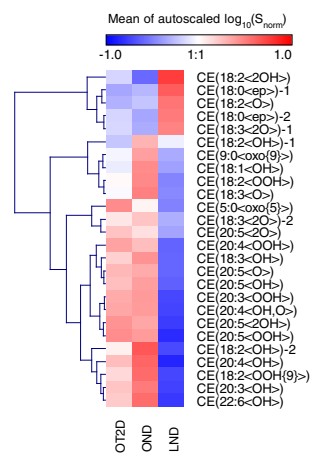 **d** 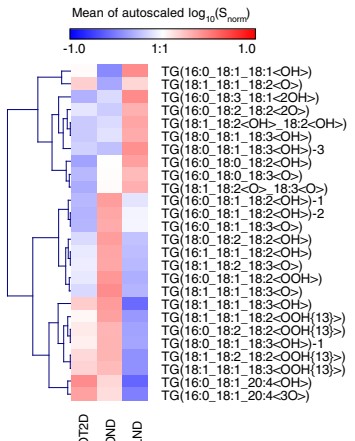 **e** 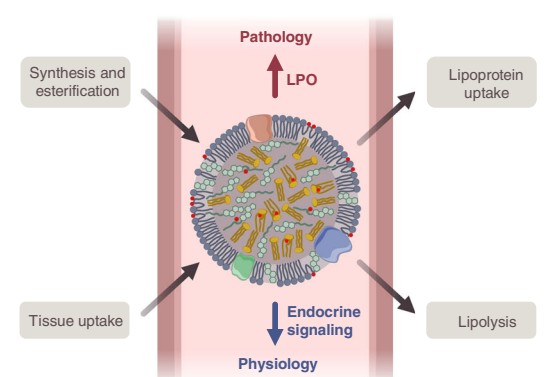

**Fig. 5 | Quantification of oxidized complex lipids in blood plasma lipid extracts of lean non-diabetic (LND), obese non-diabetic (OND) and obese with type 2 diabetes (OT2D) individuals.** In silico prediction of sample-specific epilipidome, its identification using stDDA followed by PRM-based targeted quantification allowed to annotate and quantify oxidized lipids with significant differences between three phenotypes (**a**). Heatmaps illustrate lean (LND), obese (OND) and T2D obese (OT2D) specific epilipidomics signatures for oxPC (**b**), oxTG (**c**) and oxCE (**d**) lipids with significantly different levels (ANOVA p < 0.05). Source data are provided as a Source Data file. Based on the identified phenotype-specific epilipidomes, the endocrine functionality was proposed for complex lipids esterified octadecanoids, which was altered upon obesity and type 2 diabetes via enrichment of lipid peroxidation (LPO)-derived eicosanoids (**e**).

15-lipoxygnase (LOX) (at least to some extent) in formation of these PC species. Specific enrichment of 13-HODE isomer was even more evident for CE(18:2<OOH>) (Supplementary Fig. 4e).

Again, within oxTGs with higher abundance in LND vs obese individuals, the majority was represented by hydroxy-modified octadecanoids (Fig. 5c), indicating that lipid class-independent octadecanoid accumulation is a hallmark of human plasma of lean individuals. It was previously established that de-esterified octadecanoid alcohols are elevated in TG-rich lipoproteins, especially in VLDL[4], and it is worthwhile to speculate that their TG-esterified forms enriched in LND samples represent a potential storage and/or transport mechanisms. On the other hand, oxCE elevated in LND individuals were represented by epoxy- and diol-modified octadecanoids, indicating enrichment of CYP450- and soluble epoxide hydrolase pathway-derived metabolites (Fig. 5d). Indeed, the role of octadecanoids as physiologically relevant signaling molecules emerged in several recent reports. Thus, elevated levels of linoleic- and α-linolenic acid-derived oxylipins were shown to be negatively associated with the development of type 1 diabetes[43], cardiovascular events[44], and acutely decompensated cirrhosis[45]. Esterified diols of linoleic acid in TG-rich lipoproteins formed a characteristic signature promoting anti-atherogenic endothelial phenotype[46]. However, in all those studies only oxylipins (free and released from complex lipids by basic hydrolysis) were measured. Here we demonstrated that those oxidized acyl chains are present within complex lipids of different classes with a certain specificity of modification patterns (e.g. alcohols for oxTG and epoxides/diols for oxCE) and enriched in blood plasma of LND vs obese individuals. This finding provides direct evidence for the previously proposed endocrine signaling role of oxylipins usually referred only as paracrine and autocrine molecules[4]. Indeed, oxidized complex lipids incorporated in blood lipoproteins can be delivered to or collected from different organs and tissue to support their controlled release or clearance via the coordinated action of lipoprotein lipases and acyltransferases, respectively.

The pattern of oxidized complex lipids was remarkably altered in obese individuals, with overall enrichment of non-enzymatic LPO products evident for all analyzed lipid classes. Thus, in the blood plasma of obese individuals we identified elevated levels of hydroperoxides as well as a variety of hydroxy-derivates. In this case, the majority of oxPCs contained modified eicosatetraenoic and eicosatrienoic acyl chains. Multiple positional isomers carrying hydroxyeicosatetraenoic (HETE) acyl chains including 7-, 8-, 9-, 11-, 12-, and 15-OH were enriched in the samples from obese individuals, indicating the free radical origin of these oxPCs (Fig. 5b). Thus, for PC(16:0_20:4<OH>) we quantified three groups of hydroxyeicosatetraenoic acid (HETE) containing regioisomers represented by 15-HETE, 11/12-HETE and 7/8/9-HETE species (Supplementary Fig. 4c, d). Both 11/12-HETE and 7/8/9-HETE were significantly higher 15-HETE analogues in all studied conditions. Whereas for PC(18:0_20:4<OH>), 11/12-HETE-containing species showed significantly higher intensities relative to 8/9-HETE isomers. This suggests possible

involvement of 12-LOX activity in their synthesis, in addition to LPO. Overall, the levels of LPO-derived oxPCs were higher in OT2D individuals with the prevalence of modified C20:x acyl chains, whereas in OND samples, PCs with oxidized C18:x chains were detected as well. This pattern discriminating between OND vs OT2D individuals was even more evident for oxTG (Fig. 5c), with C18:x hydroxy lipids enriched in OND, and the majority of LPO-derived hydroperoxides (both C18:x and C20:x origin) elevated in both OND and OT2D samples. The difference between OND and OT2D individuals in terms of oxCE followed the same pattern, with oxidized C18:x-containing oxCEs enriched specifically in OND, and the general LPO products derived mostly from C20:x acyl chains characteristic for both obese phenotypes (Fig. 5d). For instance, the impact of LPO was clearly evident for CE(20:4<OOH>), where 15-HETE, 12-HETE, and especially 9-HETE isomers were all abundantly present (Supplementary Fig. 4f).

Taken together, we could demonstrate that complex lipids within human blood carry a large variety of oxidized acyl chains supporting their endocrine functionality (Fig. 5e). Moreover, LND individuals have a distinct epilipidomic signature enriched in oxygenated octadecanoid acyl chains with modification type specificity between different lipid classes. This physiological signature is disrupted by the obesity-induced metabolic distress with the formation of various products, mostly of non-enzymatic origin. Both OND and OT2D individuals were characterized by this LPO-driven epilipidomics shift, however in OND a number of modified C18:x species, different from the ones present in lean individuals, was enriched indicating possible attempts towards systemic adaptation.

Finally, to understand if the changes in intensities of oxidized complex lipids might simply reflect the changes in abundance of corresponding non-oxidized lipids, we performed LC-MS/MS lipidomics analysis from the same set of samples and quantified PC, CE, and TG lipids in LND, OND, and OT2D groups. Obtained lipid quantities were correlated with intensities of oxidized lipids described above (Supplementary Fig. 5). Interestingly, although linear regression analysis was performed using "all-against-all" approach, positive correlations were obtained only in a lipid class-specific manner (e.g. TG with oxTG). The highest number of positive correlations was observed for LND sample group and the lowest for OT2D group. Within all groups, the most correlated species were represented by TG lipid class. However, despite a high number of significantly correlated species in LND group, none of the correlated oxidized lipids showed significant differences between studied groups (Fig. 5 and Supplementary Fig. 5a). In OND group, among numerous positive correlations between native and oxidized lipids, only three oxTGs and one oxCE (Fig. 5c, d, Supplementary Fig. 5b, marked with *) showed significantly higher intensities relative to LND group. OT2D group showed poor correlation in general with only three native TGs correlating with two oxTG species (Supplementary Fig. 5c). In general, despite large numbers of positively correlated native and oxidized lipids, the overall number of substrate−product pairs (native lipid and its corresponding modified form; Supplementary Fig. 5a, b, blue squares) was low, indicating selectivity of lipid modifications both in lean and obese individuals. Underlying mechanisms of this selectivity remain unclear. However, it emphasizes the significance of future research on the functional role of oxidized complex lipids in metabolic disorders.

Here we presented a comprehensive analytical and bioinformatics workflow for the accurate identification and relative quantification of oxidized complex lipids. Using in vitro generated standards we optimized MS1 (selection of preferential ionization adducts) and MS2 (lipid class-specific elevated energy HCD) methodology for the accurate annotation of oxidized complex lipids complimented with the RPC-based retention time mapping for the resolution of isomeric species and correction for in-source fragmentation. To address sample-specific epilipidome composition, blood plasma PUFA-containing lipids served as substrates for in silico prediction of oxidized

species, which in turn were used as an inclusion list for semi-targeted DDA analysis. Using the optimized LC-MS/MS workflow and defined fragmentation rules, oxidized complex lipids can be identified in modification type- and position-specific manner.

The workflow was used for comprehensive profiling and relative quantification of oxPC, oxCE and oxTG lipids in blood plasma of lean, non-diabetic and type 2 diabetic obese individuals, with over 300 modified species identified of which 116 were relatively quantified. We could illustrate significant remodeling of the blood plasma epilipidome upon the development of obesity and its complications. Epilipidomic signature characteristic to lean individuals was dominated by lipid class-specific patterns of modified octadecanoid acyl chains with an endocrine regulatory potential. Obesity and especially obesity-associated type 2 diabetes induced significant shift in the epilipidome characterized by the depletion of regulatory octadecanoid signature and the elevation of LPO-derived eicosanoids. Importantly, oxidized complex lipids correlated poorly with the non-oxidized ones, indicating high specificity of lipid modifications. This methodology opens opportunities to address the complexity of epilipidomic (dys)regulation beyond oxylipin profiling towards a comprehensive understanding of complex lipid modifications and their role as endocrine signals in pathophysiology of metabolic disorders.

## Methods

### Chemicals

Acetonitrile (MeCN), isopropanol (i-PrOH), methanol (MeOH), and formic acid (all ULC/MS-CC/SFC grade) were purchased from Biosolve (Valkenswaard, Netherlands). Chloroform (CHCl$_3$; Emsure®), methyl-tert-butyl-ether (MTBE; ≥ 99%), 3,5-di-tert-4-butylhydroxytoluol (BHT), Na-L-ascorbate, CuSO$_4$•5H$_2$O and the NIST® SRM® 1950 Metabolites in Frozen Plasma were purchased from Sigma-Aldrich (Taufkirchen, Germany). Ammonium formate (NH$_4$HCO$_2$; MS grade) was purchased from Fluka Analytical (München, Germany). Water (ddH$_2$O) was ultrapurified by an ELGA PURELAB Ultra Analytic (Berlin, Germany) instrument delivering water quality of a resistivity ≥ 18.2 MΩ•cm.

### Lipid standards

1-Palmitoyl-2-linoleoyl-sn-glycero-3-phosphocholine (PC 16:0/18:2), 1-palmitoyl-2-oleoyl-sn-glycero-3-phosphocholine (PC 16:0/18:1), and SPLASH® LIPIDOMIX® were purchased from Avanti Polar Lipids (Avanti Polar Lipids, Inc., Alabama, USA). 1,2-Dipalmitate-3-linoleate-glycerol (TG 16:0/16:0/18:2) and cholesteryl linoleate (CE 18:2) were purchased from Larodan (Solna, Sweden). Oxylipin standards (15(S)-hydroperoxyeicosatetraenoic acid, 15(S)-hydroxyeicosatetraenoic acid, 14(15)-epoxyeicosatetraenoic acid, 15-oxoeicosatetraenoic acid, 8(S),15(S)-dihydroxyeicosatetraenoic acid) were from Cayman Chemical (Ann Arbor, USA).

### In vitro oxidation of lipid standards and human blood plasma

Lipid standards (PC 16:0/18:2, TG 16:0/16:0/18:2, and CE 18:2) and human plasma were oxidized in the presence of copper sulfate and sodium ascorbate. Liposomes (PC 16:0/18:2) or micelles (TG 16:0/16:0/18:2 or CE 18:2 mixed with PC 16:0/18:1 in a molar ratio 3:1) were dried in amber vials under the stream of nitrogen and reconstituted in NH$_4$HCO$_2$ (10 mM, pH 7.5) at the final concentrations of 1.5 mM. CuSO$_4$ (final concentration 0.075 mM) and sodium ascorbate (final concentration 0.15 mM) were added to the liposomes or micelles (final concentration 1.125 mM) and incubated at 37 °C, 1300 rpm for 24 h. Lipids were diluted with i-PrOH to the final concentration of 0.15 mM prior to the analysis. Lipids from in vitro oxidized blood plasma were extracted as described below.

### Human blood plasma lipid extraction

Plasma samples (n = 150) were selected from the Leipzig Obesity Bio-Bank that has been approved by the Ethics committee of the University

of Leipzig (approval number: 159-12-21052012). All subjects gave written informed consent before contributing samples and data to the biobank. EDTA plasma was collected for all samples in a way that minimized cell activation, no vacutainers were used. Samples were always treated the same way, stored for less than 2 years, and provided for the analysis at a similar "storage age". After collection, samples were frozen and stored at −80 °C until the analysis. Donors of plasma samples were selected to compare groups of patients with obesity ($n = 100$, body mass index (BMI) 30–50 kg/m²), without (obesity, no diabetes (OND); $n = 50$) or with type 2 diabetes (obesity, type 2 diabetes (OT2D); $n = 50$), and a group of lean controls with a BMI < 25 kg/m² (lean, no diabetes (LND); $n = 50$). Data on BMI, gender, and age are provided for these three groups in Supplementary Data 9.

### Lipid extraction from human blood plasma
For lipid extraction, samples were thawed by incubating tubes containing 10 µL of plasma on ice for 1 h. SPLASH® LIPIDOMIX® was added (5 µL) and incubated on ice for 15 min. Lipids were extracted by adding ice-cold MeOH (375 µL) and MTBE (1250 µL) with subsequent vortexing[47]. Samples were incubated for 1 h at 4 °C (orbital shaker, 32 rpm). Phase separation was induced by addition of ddH$_2$O (375 µL). Samples were vortexed, incubated for 10 min at 4 °C (orbital shaker, 32 rpm), centrifuged (10 min, 4 °C, 1000 × $g$), the organic phase was collected into a new tube, and the solvent was removed *in vacuo* (Eppendorf concentrator 5301, 1 mbar). A Quality Control (QC) sample was prepared by mixing obtained lipid extracts in an equivolumetric manner. All extraction solvents contained 0.01% (w/v) BHT and were cooled on ice before use.

### In silico oxidation
Lipid standards or previously reported[40] most abundant PUFA-containing human blood plasma PCs (17 molecular species), TGs (17 molecular species) and CEs (11 molecular species) (Supplementary Data 4) were used for in silico oxidation by LPPtiger software[42]. The list of modifications included hydroperoxy, hydroxy, epoxy, and keto groups as well as truncated products. The oxidation was performed at the level 1 considering a maximum of 2 sites and a maximum of one <oxo> and one <OOH> group. Elemental composition of predicted oxidized lipids was used to compose inclusion lists considering preferential ionization adducts used for MS analysis (fomate adduct anions for oxPC, sodiated adduct cations for oxCE and oxTG).

### Chromatography
Ultra-high-performance LC (RP-UHPLC) was carried out on a Vanquish Horizon (Thermo Fisher Scientific, Germering, Germany) equipped with an Accucore C18 column (150 × 2.1 mm: 2.6 µm, 150 Å, Thermo Fisher Scientific, Sunnyvale, CA, USA). Lipids were separated by gradient elution with solvent A (MeCN/ddH$_2$O, 1:1, v/v) and B (*i*-PrOH/MeCN/ddH$_2$O, 85:10:5, v/v/v) both containing 5 mM NH$_4$HCO$_2$ and 0.1% (v/v) formic acid. The separation was performed at 50 °C with a flow rate of 0.3 mL/min using the following gradient: 0–20 min−10–86% B, 20–22 min−86–95% B, 22–26 min−95% B (isocratic), 26–26.1 min−95–10% B, 26.1–34.0 min−10% B (isocratic, column re-equilibration).

### LC-MS/MS-based identification and quantification of non-oxidized PC, CE, and TG lipids
Blood plasma lipids were identified using studied group-specific pooled samples generated by mixing equal amounts of each individual sample within the group. Lipids were diluted with *i*-PrOH to a final concentration of 0.03 µL of plasma/µL *i*-PrOH and amount corresponding to 0.15 µL plasma (5 µL) was injected onto the column. For identification, RP-UHPLC (as described above) was coupled to a Thermo Scientific Q Exactive Plus Quadrupole-Orbitrap (Thermo Fisher Scientific, Bremen, Germany) equipped with a heated electrospray (HESI) probe. Mass spectra were acquired in positive and negative modes with the

following ESI parameters: sheath gas−40 arb.units, auxiliary gas−10 arb.units, sweep gas−1 arb.units, spray voltage−3.5 kV (positive ion mode) or −2.5 kV (negative ion mode), ion transfer temperature−300 °C, S-lens RF level−35%, and aux gas heater temperature−370 °C. Data were acquired in data dependent acquisition (DDA) modes with survey scan resolution of 140,000 (at $m/z$ 200), AGC target 1e6, Maximum IT 100 ms in a scan range of $m/z$ 350–1200. Data-dependent MS2 were acquired with a resolution settings of 17,500 at 200 $m/z$, AGC target 1e5 counts, Maximum injection time (IT) 60 ms, loop count 15, isolation window 1.2 $m/z$ and stepped normalized collision energies (nCE) of 10, 20, and 30% (15, 20 and 30% for unpolar lipids). A data-dependent MS2 was triggered when an AGC target of 2e2 was reached followed by a Dynamic Exclusion for 10 s. All isotopes and charge states > 1 were excluded. All data were acquired in profile mode.

For quantification purposes, blood plasma lipid extracts were separated on a RPC as described above. MS data were acquired in Full MS mode on a Q Exactive Plus Hybrid Quadrupole- Orbitrap mass spectrometer in the positive ion mode at the resolution of 140,000 at $m/z$ 200, AGC target of 1e6 and a Maximum IT of 100 ms in the mass range of $m/z$ 350–1200. Data were acquired in profile mode.

### Mass spectrometry method development for analysis of oxidized PC, CE and TG lipids
Method development and optimization for analysis of oxidized complex lipids were performed using RP-UHPLC coupled on-line either to a Thermo Scientific Q Exactive Plus Quadrupole-Orbitrap (Thermo Fisher Scientific, Bremen, Germany) or Orbitrap Fusion Lumos Tribrid (Thermo Fisher Scientific, San Jose, USA) mass spectrometers.

Q Exactive Plus Quadrupole-Orbitrap was equipped with a HESI source and operated in both positive and negative ion modes with the following parameters: sheath gas−40 arb.units, auxiliary gas−10 arb.units, sweep gas−1 arb.units, spray voltage−+3.5 kV and −2.5 kV, capillary temperature−300 °C, S-lens RF level−35%, and aux gas heater temperature−370 °C.

Orbitrap Fusion Lumos Tribrid mass spectrometer using a HESI source was operated in both positive and negative ion modes with the following parameters: spray voltage−+3.5 kV and −2.8 kV, ion transfer tube temperature−250 °C, sheath gas−25 arb.units, aux gas−10 arb.units, vaporizer temperature−200 °C, RF level−25%.

### Multistage ion activation (MS3) for analysis of in vitro oxidized standards and human blood plasma
Multistage ion activation experiments were performed on an Orbitrap Fusion Lumos Tribrid mass spectrometer using inclusion list DDA mode. Full MS was acquired at the resolution 120,000 at $m/z$ 200, scan range of $m/z$ 450–1000, AGC target 4e5 counts, Maximum IT 100 ms. MS/MS events were triggered for precursor ions from the inclusion lists. MS2 spectra were acquired at the resolution 15,000 at $m/z$ 200, AGC target 8e4, Maximum IT 100 ms, isolation window 1 $m/z$, nCE 25% (negative mode) and 37% (positive mode). The filters used were MIPS (small molecule), charge state (1), dynamic exclusion (exclusion duration 6 s, mass tolerance ± 10 ppm), target exclusion (polarity-specific) and an inclusion list (class-specific).

MS3 analysis was conducted using the orbitrap at the resolution setting of 15,000 for $m/z$ 200, AGC target 1e5, Maximum IT 250 ms, injection ions for all available parallelizable time, isolation window 2 $m/z$, nCE 33%. The following filters were used: precursor selection range (100–450 $m/z$); targeted exclusion mass list (including non-oxidized fatty acid list, mass tolerance ± 50 ppm); product ion trigger (including non-oxidized fatty acid list, mass tolerance ± 25 ppm).

### Semi-targeted data-dependent acquisition (stDDA) for oxidized lipids identification
stDDA analysis was performed on a Q Exactive Plus Quadrupole-Orbitrap mass spectrometer. Full MS spectra were acquired at the

resolution of 140,000 at $m/z$ 200, scan range 380–1200 $m/z$ (negative ion mode, 0–17 min) and 380–1200 $m/z$ (positive ion mode, 17–34 min), AGC target 1e6 counts, Maximum IT 100 ms. MS/MS events (top 6) were triggered for precursor ions from the inclusion lists (in silico oxidized PC in negative, and oxCE and oxTG in positive mode). MS/MS spectra were acquired at the resolution of 17,500 at $m/z$ 200, AGC target 1e5, Maximum IT 200 ms, loop count 6, isolation window 1.2 $m/z$, fixed first mass 100.0 $m/z$, stepped nCE 20-30-40% (negative mode) and 30-40-50% (positive mode). The filters used were minimum AGC target (1e1), charge state (1), isotope exclusion (on), apex trigger (up to 6 s), and dynamic exclusion (3 s). All spectra were acquired in profile mode.

### Targeted parallel reaction monitoring (PRM) for oxidized lipids relative quantification

For PRM, inclusion lists were used in retention time-scheduled negative (0–17 min) and positive modes (17–34 min). MS/MS spectra were acquired at the resolution of 17,500 at $m/z$ 200, AGC target 2e5 counts, Maximum IT 200 ms, isolation window 1.2 $m/z$, stepped nCE 20-30-40% (negative mode) and 30-40-50% (positive mode).

### Identification and quantification of non-oxidized lipids

DDA LC-MS/MS datasets were used for lipid species identification. Lipostar v. 1.0.6 (Molecular Discovery, Hertfordshire, UK) equipped with LIPID MAPS structure database (version December 2017) was used. The raw files were imported directly and aligned using default settings. Automatic peak picking was performed with SDA smoothing level set to low and minimum S/N ratio 3. Automatic isotope clustering settings were set to 7 ppm with RT tolerance 0.2 min. An "MS2 only" filter was applied to keep only features with MS/MS spectra for identification. Following parameters were used for lipid identification: 5 ppm precursor ion mass tolerance and 10 ppm product ion mass tolerance. The automatic approval was performed to keep structures with quality of 3-4 stars.

Quantification of non-oxidized PC, CE, and TG lipids was performed using Skyline v. 21.1.0.146 (MacCoss Lab, University of Washington, Seattle, USA)[48]. For each lipid, corresponding precursor ion was selected for the peak integration. The peak boundaries were delimited, manually corrected and verified. The obtained peak areas were normalized by appropriate lipid species from SPLASH® Lipidomix Mass Spec Standard (Avanti): PC 15:0/18:1(d7) for PC, CE 18:1(d7) and TG 15:0/18:1(d7)/15:0 for CE and TG, correspondingly. Type I isotopic correction as well as correction for the incomplete labeling of deuterated ISTDs were applied as described previously[49,50].

### Identification and quantification of oxidized lipids

Oxidized lipids were identified by LPPtiger 2.0. LPPtiger 2.0 is a major update of the previously published LPPtiger software[42] aiming to extend the identification of oxidized GPLs to additional lipid classes. LPPtiger 2.0 extends the in silico epilipidome predictions, in silico fragmentation, and identification to cover oxTG, oxDG, and oxCE lipids. The overall workflow has been optimized to merge the processing of different lipid subclasses into a single task while keeping the initial algorithms so that at least 10 times speed improvement was achieved. LPPtiger 2.0 includes Inclusion List Generator to create inclusion lists for preparation of targeted acquisition methods. The exported lists include precursor $m/z$ as well as $m/z$ of fragment ions corresponding to oxidized fatty acyl chains which can be used directly to design PRM and MRM experiments. For the identification of oxidized lipids, mzML files converted from polarity switching mode are now supported in LPPtiger 2.0. Moreover, LipidLynxX[51] is directly integrated into LPPtiger 2.0 to perform lipid nomenclature conversions. In silico predicted epilipidome can be exported by LPPtiger 2.0 into JSON format including structured information of elemental composition, fragmentation pattern, corresponding structures in SMILES format, along with other predictions such as fingerprint $m/z$

lists. In addition to the XLSX identification table and interactive HTML report with six-panel-figure, the identification results can also be dumped into a JSON file including all spectra assignments and scoring details. These exported JSON files provide fully transparent access to all predictions and identification details and enable further advanced post processing by a data scientist. LPPtiger 2.0 is a cross platform (Linux, macOS, and Windows) open-source software released under AGPL license for academic users, the source code and executable files for Windows platform are freely available on GitHub repository: https://github.com/LMAI-TUD/lpptiger2.

For manual control of identification accuracy, and for detailed assignment of modification type- and position-specific fragment ion signals, lists of in silico oxidized lipids generated by LPPtiger 2.0 with possible "precursor-oxidized fatty acid (oxFA) fragment" pairs as well as fragmentation rules defined for the in vitro oxidized standards (Supplementary Data 2) were used as a support. All MS/MS spectra were screened in Qual Browser, Thermo Xcalibur v. 4.2.47 (Thermo Fisher Scientific Inc.).

Quantification of oxidized lipids was performed using PRM-derived data in Skyline v. 21.1.0.146 (MacCoss Lab, University of Washington, Seattle, USA)[48]. For each targeted precursor, the oxFA fragment ion was selected for the peak integration. The peak boundaries were delimited, manually corrected and verified. The obtained peak areas were normalized by appropriate lipid species from SPLASH® Lipidomix Mass Spec Standard (Avanti): LPC 18:1(d7) for truncated oxPC, PC 15:0/18:1(d7) for full-length oxPC, CE 18:1(d7) and TG 15:0/18:1(d7)/15:0 for oxCE and oxTG, correspondingly.

### Data analysis and visualization

For RT mapping, hydrogen-based Kendrick mass defects (KMD(H)) were calculated for each identified lipid and plotted against RT using R-scripts available at https://github.com/SysMedOs/AdipoAtlasScripts/tree/main/DataVisualization[49].

Quantified normalized peak areas were median-centered, log-transformed, and autoscaled in MetaboAnalyst v.5 (https://www.metaboanalyst.ca, Xia Lab, McGill University, Montreal, Canada)[52]. Data processed by MetaboAnalyst were exported as CSV files and used in other software for visualization.

Oxidized lipids with statistically significant difference were selected for preparing heatmaps. The latter were created in Genesis v.1.8.1 (Thallinger Lab, Graz University of Technology, Graz, Austria)[53]. Features (oxidized lipids) were clustered by average linkage weighted pair group method with arithmetic mean (WPGMA) agglomeration rule. Other graphs were created in OriginPro 2019 v. 9.6.0.172 Academic (OriginLab Corporation, Northampton, USA).

For correlation analysis between non-oxidized and oxidized PC, CE, and TG lipids, concentrations of non-oxidized and relative intensities of oxidized lipids were used. The potential correlation between each detected native lipid and each detected oxidized lipid was evaluated by simple linear regression using an "all-against-all" approach. In addition to the coefficient of determination ($R^2$), the predictive relevance ($Q^2$) was evaluated using the leave-one-out cross-validation. Only correlations having $R^2 \geq 0.5$ and $R^2-Q^2 \leq 0.2$ were selected for inspection and used to generate a bubble plot in OriginPro 2019 v. 9.6.0.172 Academic (OriginLab Corporation, Northampton, USA).

### Reporting summary

Further information on research design is available in the Nature Research Reporting Summary linked to this article.

## Data availability

Data generated during the study (all raw LC-MS/MS files) available in a public repository MassIVE MSV000088608 [https://doi.org/10.25345/C5SG5C]. Supplementary Data 11 describes which file names correspond to which LC-MS/MS datasets. Source data are provided with this paper.

For lipid identification, the LIPID MAPS structural database (version December 2017) was used. Source data are provided with this paper.

## Code availability

LPPtiger 2.0 is a cross platform (Linux, macOS, and Windows) open-source software released under AGPL license for academic users, the source code and executable files for Windows platform are freely available on GitHub repository: https://github.com/LMAI-TUD/lpptiger2.

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

## Acknowledgements

Financial support from the German Federal Ministry of Education and Research (BMBF) within the framework of the e:Med research and funding concept for SysMedOS project (to MF), "Sonderzuweisung zur Unterstützung profilbestimmender Struktureinheiten 2021" by the SMWK, and Deutsche Forschungsgemeinschaft (FE 1236/5-1 to MF) are gratefully acknowledged. This publication is based upon work from COST Action EpiLipidNET, Pan-European Network in Lipidomics and Epilipidomics (CA19105; https://www.epilipid.net), supported by COST (European Cooperation in Science and Technology). We thank Prof. Ralf Hoffmann (Institute of Bioanalytical Chemistry, University of Leipzig) for providing access to his laboratory. We are grateful to Ken Cook (Thermo Fisher Scientific, UK) for manuscript proofreading.

## Author contributions

M.F. conceived the project, guided the research, assisted with the experiments and data interpretation, and wrote the manuscript. A.C. designed and performed most of the experiments, analyzed and interpreted data. P.N. performed oxidized lipid identification and most of the data analysis, validation, and visualization. M.L. prepared blood plasma lipid extracts. D.G. performed stDDA, PRM analysis, and oxidized lipid identification. Z.N. programmed LPPtiger 2.0 version. M. Blüher provided human blood plasma samples. M. Baroni, G.C., and L.G. performed statistical data analysis. All authors edited and approved the manuscript.

## Funding

## Competing interests

M.B. received honoraria as a consultant and speaker from Amgen, AstraZeneca, Bayer, Boehringer-Ingelheim, Lilly, Novo Nordisk, Novartis and Sanofi. The remaining authors declare no competing interests.

## Additional information

[1]Center of Membrane Biochemistry and Lipid Research, Faculty of Medicine Carl Gustav Carus of TU Dresden, 01307 Dresden, Germany. [2]Institute of Bioanalytical Chemistry, Faculty of Chemistry and Mineralogy, University of Leipzig, 04013 Leipzig, Germany. [3]Center for Biotechnology and Biomedicine, University of Leipzig, 04013 Leipzig, Germany. [4]Thermo Fisher Scientific, 63303 Dreieich, Germany. [5]Molecular Discovery, Kinetic Business Centre, Borehamwood WD6 4PJ Hertfordshire, UK. [6]Department of Chemistry, Biology and Biotechnology, University of Perugia, 06123 Perugia, Italy. [7]Medical Department III (Endocrinology, Nephrology and Rheumatology), University of Leipzig, 04103 Leipzig, Germany. [8]Helmholtz Institute for Metabolic, Obesity and Vascular Research (HI-MAG) of the Helmholtz Zentrum München at the University of Leipzig and University Hospital Leipzig, 04103 Leipzig, Germany. [9]These authors contributed equally: Angela Criscuolo, Palina Nepachalovich. ✉e-mail: maria.fedorova@tu-dresden.de

