## [Peer Review File · Nature Communications]

REVIEWER COMMENTS

Reviewer #1 (Remarks to the Author):

Criscuolo et al. describe the development of a methodology to detect oxidized lipid derived from glycerophospholipid and neutral lipids using bioinformatics approaches and LC-MS/MS technologies. First, they systematically defined the ionization preferences and fragmentation rules of oxidized lipid derived from glycerophospholipid and neutral lipids. Based on a developed methodology, they revealed the characteristic epilipidomic profiles in lean and obese individuals. The concept and the general approach are of great interest, but a number of issues need to be addressed in order for the publication to be accepted. Overall, I would recommend major revisions. See below for detailed comments.

- Authors mentioned “Liquid chromatography coupled on-line to mass spectrometry (LC-MS) was actively applied for the identification and quantification of oxidized complex lipids with a main focus on oxidized phosphatidylcholines (oxPC) and oxCE.”. However, there are many studies on the structural analysis of oxidized-phosphatidylethanolamines (<https://www.nature.com/articles/nchembio.2238>), -phosphatidylserines (<https://www.nature.com/articles/s41589-018-0195-0>) and -triacylglycerol (<https://www.nature.com/articles/s41467-017-02186-9>).

- The use of PL instead of GPL is an oversight. PL would include sphingomyelin (SM) but GPL would not. GPL commonly contain PUFAs whereas SM does not. It is advised to use GPL instead of PL. Also, authors described PC as being diacyl which is certainly true yet PCs and GPL in general have a portion of their structures present in alkyl and/or vinyl ether. The presence of oxidized ether GPL is not discussed.

- Authors showed that oxCE and oxTG tend to form sodiated adducts. However, there is no sodium ions in LCMS mobile phases described in Method section. Where were the sodium ions provided from?

- Do you have any indication why the ionization preferences differ between non-oxidized or oxidized CE/TG?

- Authors claimed “the general trend was defined as well with earlier elution times for the species carrying the modification closer to the ω -end of the acyl chains.”. Does this rule work for more complex oxidized lipids, such as mono-oxidized arachidonate-PC?

- As authors mentioned, lipid peroxidation leads to the formation of numerous isomeric species. Is it possible to map the retention times of all these complex isomers, including truncated species? Authors need to clearly mention the limitations of their developed methodology.

- To the best of my knowledge, author have previously developed “LPPTiger” as a tool for identification of oxidized glycerophospholipids (Ni Z et al. Sci. Rep. 2017), not oxidized neutral lipids. Are there any functional improvements in LPPTiger used in this study? The authors need to provide description of these.

- While this study profiles oxPC, oxCE and oxTG generated in lean and obese individuals, the study is rather descriptive in nature and it is unclear how this information would help with design of treatments or other preventative strategies, or how oxidized lipids are involved in form of obesity or diabetes.

- What is the abundance of non-oxidized PC, CE and TG in individual serum samples? What is the extent of oxidation for individual lipids?

- Several oxidized lipids were observed in serum samples, such as mono-oxidized and hydroperoxidized lipids. However, many of them are well known oxidized lipids and can be detected by conventional LC/MS/MS analysis used in previous studies. The advantage of using a newly developed method is not clear.

Reviewer #2 (Remarks to the Author):

Criscuolo and colleagues describe an LC-MS/MS approach to quantify and identify oxidized complex lipids in a semi-targeted manner. They apply this approach to a comparison of plasma from individuals varying in obesity and diabetic status. This work addresses a gap in current lipidomics approaches as the measurement of oxidized lipid species is often neglected due to their specific challenges including their often low abundance. As the authors state correctly, however, there is much reason to believe that the levels of oxidized lipids provide a key readout for many biological

conditions. It is a particular challenge with current approaches to identify the location and type of oxidation. Work towards a streamlined workflow for measuring oxidized lipids that can be applied to a variety of biological questions is thus highly needed.

The authors did a great job in comprehensively annotating a large number of fragmentation spectra from oxidized PC, CE, and TG lipids and making them available for future readers (e. g. in Figure 1 and in supplementary Files S1, S2, S3, and S4) and I consider this work to be the heart of the manuscript, alongside showcasing how this information can be used to measure their abundances in biological samples.

As I judge the topic of the manuscript to be highly relevant and the execution and interpretation of the LC-MS/MS to be highly skilled and detailed, I advocate for clarifying the method to increase applicability and thereby impact. My main suggestion is thus to provide the exact methodology that the reader shall implement when aiming to apply it to their own samples. For large parts, the manuscript remains somewhat elusive as to what exactly the 'developed methodology' and the 'holistic workflow' entails. It is unclear what is meant by 'biological context-specific' profiling (as e. g. on p. 4) and it should be clarified why and at what stage samples should be pooled (p. 12). Figure 4A, for example, does not mention pooling of samples. Which extraction method and mobile phases do the authors recommend for different targets, especially in light of the variation with adduct type? Does this need to be optimized for each lipid class? Similarly, do the stepped CE values for the elevated energy HCD need to be adjusted? Does the LPPTiger software generate the inclusion list for you?

A prior publication by the same group (reference 39) details the bioinformatic approach that "allows for the prediction of sample-specific oxidized lipidome" which is presumably what 'in silico prediction' (e. g. p. 10) refers to. I laude the upload of (part of the) code to github but unfortunately and while the authors mention 'high-throughput identification' (p. 9), the data analysis and visualization strategy overall seems rather fragmented, using a number of software tools (LPPTiger, QualBrowser, Skyline, R, MetaboAnalyst etc.). This may exceed the scope of this publication, but I advocate for including as many steps as possible within GUI-based software for optimal reproducibility and dissemination. Perhaps a LPPTiger version 2 could include retention time mapping, KMD plots, etc. Along with that, do you recommend potential users to do a manual inspection as detailed on p. 8 for every new sample type and lipid class? I further wonder, whether the authors could comment on a possible implementation of this knowledge as on-line intelligent data acquisition? If not, what are the bottlenecks?

Lastly, I am not an expert on the biological example chosen and thus refrain from commenting in this regard. However, the number and extent of observed changes in the different groups is promising. A comparison to the direction and extent of changes observed in the corresponding non-oxidized species may add additional biological insights as well as showcase the need for measurements of

oxidized complex lipids. Similar, could the same conclusions have been drawn based on the oxylipin levels alone? Since the authors collected untargeted data, these comparisons may be possible without the need for additional data collection. Are there any other quantified features that show significant differences between the groups, possibly even track with the oxidized lipids?

Additional minor comments:

Presumably to circumvent the lack of available standards for oxidized complex lipids, the authors chose an 'in vitro oxidation' of PC lipid standards. While this certainly seems to be a viable approach, I am curious whether the authors could comment on how well this reflects naturally occurring oxidized lipids. Since we agree that one of the main challenges is chemical diversity of these lipids and that they may be partly enzymatically modified, I am wondering if this approach does not categorically exclude a subset of complex oxidized lipids that is not reproduced by the in vitro oxidation. In addition, am I assuming correctly that besides the MS analysis, we have no data to confirm the identity of these in vitro oxidized standards? In line with that, when the authors mention 'clean' spectra (p. 8), how can we be sure that they are produced by just one isomer?

I may suggest a title that either focuses on the methodological approach or reflects the findings of applying this approach to the biological question. It is hereby of question whether the presented method should be referred to as 'holistic' when currently exclusively covering the three lipid classes of PC, CE, and TG lipids. Even if perhaps 'holistic' is referring to the depth of identification, I would suggest clarification of the terminology.

It may be advisable to refrain from using the term 'healthy' when referring to non-obese individuals (as on p. 12), not only because of the ongoing debate about obesity as a disease but also because of the potential confusion with diabetic status. From the methods and Table S4, it is unclear whether 'healthy' is describing anything beyond non-obese and non-diabetic.

For Figure 1C, S3 and S4, the authors include NL as a measure of absolute intensity, for the sake of completion and comparability it would be good to also include this for oxPCs (in Figure 1 and File S2). According to the caption, Figure 1A is supposed to show an MS3, however, in the figure it reads MS2 - this should be double-checked. As mentioned, a balance between absolute intensity and informative fragmentation needs to be struck, thus the reader would appreciate a measure of absolute intensity for the comparison of MS3 vs MS2 fragmentation.

In Figure 1E, the authors show the average abundances for 10 oxidized lipid species - is there a measure for how similar they behave, such as a standard deviation?

The authors impressively show the impact of ISF, e. g. in Figure 3D. Perhaps a supplementary Figure could show a comparison in absolute intensity, so that one may gather the true impact. After identifying occurring ISF, did you alter the source settings to minimize it? If yes, which are the recommended settings? In Figure 3A, is smoothing applied? If trying to show chromatographic performance, it may be more appropriate to not, or at least mention whichever it is. In Figure 3B, the false positive IDs should be labeled as such.

The table in Figure 4B shows that about $\frac{1}{3}$ of predicted lipids were identified. It would be interesting to know how many MS2s were triggered, i. e. the success rate of the identification. Is overlap in chromatography or quality of MS2s the main limiting factor? How close are we to a holistic identification? How often was the initial identification correct compared to after manual inspection (p. 12)? Is there the potential to assign a score for future automation?

In the present manuscript, 17 PC, 17 TG, and 11 CE species were chosen as substrates for oxidation prediction. Is this the current limitation? I wonder if the authors could comment or speculate on what the theoretical or practical limit for this type of analysis is.

Lastly, the manuscript and the clarity of its message would benefit from editing by an English speaker of native or native-like command and page and line numbers would make it easier for reviewers to comment precisely.

Reviewer #3 (Remarks to the Author):

Here, Maria Fedorova and colleagues use mass spectrometry tools that they have developed in house to predict the likely oxygenated complex lipids that maybe present in plasma, and then they use this information to screen for changes in a large number of lipids in patient sample from healthy, obese and obese insulin resistant individuals. They find significant differences that could be very useful in terms of pushing forward what we know about lipid oxygenation in the circulation.

The study is conducted to a very high degree of technical accuracy, and the authors should be congratulated on this.

I have three comments on the data itself.

1. The naming of the CE 18:1 epoxy... chemically, the convention is that the number coming after the colon includes the combined number of both rings and double bonds, not just double bonds. Thus, this should really be called 18:2 in line with the shorthand nomenclature of Liebisch et al.

2. What form of plasma was used, EDTA, citrate or heparin? Was it taken in a way to minimise cell activation? If vacutainers were used, then activation of white cells and platelets can be a significant problem. How old was it? In storage for many years, oxidation can still happen, even at -80, so were all samples of a similar age in the cohort?

3. The heatmaps of plasma data are shown as log₁₀ of normalised data, meaning that the reader can't make any easy comparisons between different lipids within the same samples, e.g. it isn't possible to tell which are more abundant than others. This is important since if the -OH forms of PE were primarily 12-HETE, or 5-HETE, then some speculation on their origin could be included. e.g. they may be considered to originate from platelets, or white cells and then perhaps they are present in extracellular vesicles instead of lipoproteins. On the other hand if there is an equal amount of all positional isomers, then this looks more like it could be driven by non-enzymatic oxidation. Hydrolysis and analysis of FA side chains for S/R ratios can also help but this may not be required if the relative abundance of positional isomers was known. This also applies for 9 and 13-HODE, in the case of LOXs and the CE isomers detected.

4. Moving forward from above, it would be very good if the data could be further interpreted biologically to ask (i) were enzymes involved or is this non-enzymatic, and (ii) where are the lipids, are they in lipoproteins, or cell derived EVs? I suspect that there maybe a different origin of the phospholipids (but perhaps the TG and CE are all in lipoproteins) due to the different FA composition that was seen.

5. In our hands, lipid hydroperoxides are very unstable when trying to analyse them from actual biological samples. While we can extract them from buffer and measure them using LC/MS/MS very easily, we find that as soon as they are added to biological samples like disrupted cells, then they decompose rapidly (likely due to metal dependent redox cycling). Did the authors conduct controls to ensure that this was not a problem in their system and how did they minimise this? Were chelators added to prevent this?

Response to the Reviewers

We are very thankful to all the Reviewers for constructive questions and comments which allowed us not only to improved significantly the presentation of our new methodology and obtained results, but also promoted us to perform a set of new experiments, which resulted in novel findings and further elaborated on the significance of oxidized complex lipids in physiological conditions and remodelling of epilipidome upon development of obesity and type 2 diabetes.

Reviewer #1 (Remarks to the Author):

Criscuolo et al. describe the development of a methodology to detect oxidized lipid derived from glycerophospholipid and neutral lipids using bioinformatics approaches and LC-MS/MS technologies. First, they systematically defined the ionization preferences and fragmentation rules of oxidized lipid derived from glycerophospholipid and neutral lipids. Based on a developed methodology, they revealed the characteristic epilipidomic profiles in lean and obese individuals. The concept and the general approach are of great interest, but a number of issues need to be addressed in order for the publication to be accepted. Overall, I would recommend major revisions. See below for detailed comments.

- Authors mentioned “Liquid chromatography coupled on-line to mass spectrometry (LC-MS) was actively applied for the identification and quantification of oxidized complex lipids with a main focus on oxidized phosphatidylcholines (oxPC) and oxCE.”. However, there are many studies on the structural analysis of oxidized-phosphatidylethanolamines (<https://www.nature.com/articles/nchembio.2238>), -phosphatidylserines (<https://www.nature.com/articles/s41589-018-0195-0>) and -triacylglycerol (<https://www.nature.com/articles/s41467-017-02186-9>).

We included above mentioned references in the revised version of the manuscript to illustrate that other complex oxidized lipids were detected in variety of biological samples as well.

- The use of PL instead of GPL is an oversight. PL would include sphingomyelin (SM) but GPL would not. GPL commonly contain PUFAs whereas SM does not. It is advised to use GPL instead of PL. Also, authors described PC as being diacyl which is certainly true yet PCs and GPL in general have a portion of their structures present in alkyl and/or vinyl ether. The presence of oxidized ether GPL is not discussed.

We substituted PL to GPL within the text of the revised version of the manuscript.

Indeed, we looked for ether forms of PC lipids as well (see Table S3 for the list of native lipids used for *in silico* oxidation). However, none of the *in silico* predicted forms of ether PC were identified in blood plasma lipids extracts in this study.

- Authors showed that oxCE and oxTG tend to form sodiated adducts. However, there is no sodium ions in LCMS mobile phases described in Method section. Where were the sodium ions provided from? • Do you have any indication why the ionization preferences differ between non-oxidized or oxidized CE/TG?

Sodium ion is one of the most abundant cations in the natural environment including biological matrix itself, but also glassware and solvents. Thus, presence of Na cations in solution allows new adduct chemistry for oxidized lipids which can be explained by the principles of physical organic chemistry. First of all, sites of cation adduction are different for non-oxidized and oxidized CE/TG lipids. For non-oxidized species it is likely an ester group attracting cations by ion-dipole interactions. Whereas upon oxidation, an oxygen modification adjacent to olefinic

motif presents a new site for cation coordination. In this case, both ion-dipole ($\text{Na}^+\dots\text{O}$) and pi-interactions ($\text{Na}^+\dots\pi$ electrons of double bonds) become possible. Generally, oxidation sites in CE/TG lipids represent three types of structures:

- Hydroperoxy and hydroxy derivatives in which O is vicinal to the conjugated olefinic motif. Here the ion-dipole $\text{Na}^+\dots\text{O}$ attraction is additionally facilitated by pi-interaction of Na^+ with conjugated double bond motif, thus the sodium adducts are efficiently formed.
- Epoxy derivatives in which O modification is adjacent to non-conjugated olefinic motif and separated by methylene group. Here, only the ion-dipole interaction is possible, so epoxy-modified CE/TG form sodiated adducts less efficiently compared to OOH/OH-modified species
- Oxo derivatives in which oxo group is vicinal to the conjugated olefinic motif. This motif represents a fully conjugated system with relatively high proton affinity. In this case, protonation of the oxo functionality is stabilized by resonance, or mesomerism. The resonance stabilization of protonated oxoCE/TG lowers the ion energy more than combined ion-dipole and pi-interactions for sodiated adducts, and this results in protonation preference over sodiation for these species.

• Authors claimed “the general trend was defined as well with earlier elution times for the species carrying the modification closer to the ω -end of the acyl chains.”. Does this rule work for more complex oxidized lipids, such as mono-oxidized arachidonate-PC?

Elution order of positional isomers of CE-OOH, e.g. CE(18:2<OOH{13}>) with RT 21.8 min vs CE(18:2<OOH{9}>) with RT 21.9 min are already illustrated at Figure 3A. In revised version of the manuscript we additionally plotted extracted ion chromatograms for mono-oxidized arachidonate-PC represented by species carrying OH group at positions 15, 13, 12, 11, 9, 8, 7, or 5 (new Figure S1). Indeed, our study has shown the following elution order for PC(16:0_20:4<OH>) positional isomers: 13<15<11<12<7<(8 and 9)<5. The position of polar modification within the acyl chain alone doesn't fully define the polarity of the solute. There are also other physicochemical factors, such as geometry of the molecule. According to Levison et al. (reference 30 in the revised version of the manuscript), HETE regioisomers elute in RPC system as following: 15<11<12<8<9<5. This is quite consistent with what we obtained.

• As authors mentioned, lipid peroxidation leads to the formation of numerous isomeric species. Is it possible to map the retention times of all these complex isomers, including truncated species? Authors need to clearly mention the limitations of their developed methodology.

As mentioned above RPC system used in this study provided sufficient resolution to distinguish many positional isomers (e.g. (CE(18:2<OOH{13}>) RT 21.8 min vs CE(18:2<OOH{9}>) RT 21.9 min). We also added new supplementary Figure S1 which illustrated elution profile of mono-oxygenated arachidonate-PC PC(16:0_20:4<OH>) positional isomers. Although baseline separation of positional isomers using extracted ion chromatograms for precursor ion was not possible for the majority of the species, however using extracted chromatograms of position specific fragment ions allowed quite accurate assignment of corresponding retention times. This statement is now included in the revised version of the manuscript as well as Figure S1 to underline the limitation of the used technology. Moreover, using this approach we quantified relative abundances for several oxPC and oxCE positional isomers. For more details, please look our reply to the Reviewer 3 comments and new supplementary Figure S4.

Separation of truncated species from oxygenated, is easily achieved. Moreover, even separation of isomeric truncated species between each other showed to be more efficient, due to the lower number of isomeric species and larger changes in solute polarities due to the structural differences. For details, please refer to the supplementary Tables S1,2 and S4 where retention times of all identified oxidized lipids are listed.

· To the best of my knowledge, author have previously developed “LPPTiger” as a tool for identification of oxidized glycerophospholipids (Ni Z et al. Sci. Rep. 2017), not oxidized neutral lipids. Are there any functional improvements in LPPTiger used in this study? The authors need to provide description of these.

Previous version of LPPTiger was significantly updated to support data processing within this study. However, majority of the new functions were only available with in-house customized LPPTiger source code through command line. To support the utility of the software for academic community, for the revised version of this manuscript we packed the source code into executable files version supported by graphical user interface, so that researchers without special bioinformatics expertise can freely use the software.

LPPTiger 2.0 is now released and can be accessed via GitHub repository: <https://github.com/LMAI-TUD/lpptiger2>. The paragraph briefly describing new functionality of LPPTiger 2.0 is now included in the corresponding Methods section. Full user manual for LPPTiger 2.0 can be found at GitHub.

· While this study profiles oxPC, oxCE and oxTG generated in lean and obese individuals, the study is rather descriptive in nature and it is unclear how this information would help with design of treatments or other preventative strategies, or how oxidized lipids are involved in form of obesity or diabetes.

The aim of the presented study was to describe the new methodology for identification and quantification of oxidized complex lipids rather than to provide treatment and preventive strategies for obesity and diabetes. However, obtained results already illustrate high specificity of epilipidomics patterns between studied groups. Indeed, presented methodology provided access to previously unknown signatures of modified complex lipids especially in lean individuals. In comparison to commonly accepted autocrine and paracrine action of oxylipins (oxygenated free fatty acids), enrichment of specific oxidized complex lipids in particular metabolic conditions allows to propose their new endocrine functionality. Furthermore, considering previously published data reporting that 90 to 95% of blood plasma oxylipins being present in circulation in esterified form, obtained results promote new investigations on the sites of formation and mechanisms of release of these bioactive molecules. For instance, based on the observations enabled by presented methodology we are planning to investigate distribution of oxidized lipids in different lipoprotein fractions in lean, obese and T2D individuals to define “donor – acceptor” relationships between different tissues in physiological and pathological conditions.

· What is the abundance of non-oxidized PC, CE and TG in individual serum samples? What is the extent of oxidation for individual lipids?

To address this question we performed a new set of experiments in which blood plasma lipid extracts prepared as before were used for identification of non-oxidized PC, CE, and TG lipids followed by quantitative lipidomics analysis in the same sample set of 150 lean, obese and obese with type 2 diabetes individuals. Obtained dataset was used for a correlation analysis against dataset featuring relative intensities of oxidized species. Details on LC-MS/MS methodologies used to acquire additional dataset are provided in Methods section of the

revised manuscript together with data processing strategies (identification, quantification, statistical analysis). To illustrate our findings, a new Figure S5 is added to the revised version of the manuscript together with a short description of the results. Briefly, this correlation analysis clearly demonstrates that abundances of oxidized lipids both in lean, obese, and especially obese with type 2 samples do not correlated with corresponding non-modified lipids.

- Several oxidized lipids were observed in serum samples, such as mono-oxidized and hydroperoxidized lipids. However, many of them are well known oxidized lipids and can be detected by conventional LC/MS/MS analysis used in previous studies. The advantage of using a newly developed method is not clear.

Indeed, some of the modified lipids reported in this manuscript were previously detected using conventional LC-MS/MS approaches. But the exact idea of the provided workflow is to obtain comprehensive representation of blood plasma epilipidome rather than few most abundant oxidized species obtained by random sampling. The advantage of this newly developed method is also a workflow aiming accurate identification of a variety of modified lipid species including modification type and position specific isomers. We believe that information provided within the manuscript (e.g. careful characterization fragmentation patterns, considerations taken to avoid false positive identifications due to in-source fragmentation, ionization preferences, elution order, etc.) together with overall workflow addressing characterization of sample specific epilipidome will allow many groups interested in oxidized lipids to advance their research. To this end, we provided all experimental details, obtained data as well as software tools (LPPtiger 2.0) with a full transparency and open access. The clear advantage of this study is identification and quantification of a set of oxidized lipids with differential abundances in lean and obese individuals at the level of molecular species with specific set of modification types including even positional isomers.

Reviewer #2 (Remarks to the Author):

Criscuolo and colleagues describe an LC-MS/MS approach to quantify and identify oxidized complex lipids in a semi-targeted manner. They apply this approach to a comparison of plasma from individuals varying in obesity and diabetic status. This work addresses a gap in current lipidomics approaches as the measurement of oxidized lipid species is often neglected due to their specific challenges including their often low abundance. As the authors state correctly, however, there is much reason to believe that the levels of oxidized lipids provide a key readout for many biological conditions. It is a particular challenge with current approaches to identify the location and type of oxidation. Work towards a streamlined workflow for measuring oxidized lipids that can be applied to a variety of biological questions is thus highly needed.

The authors did a great job in comprehensively annotating a large number of fragmentation spectra from oxidized PC, CE, and TG lipids and making them available for future readers (e.g. in Figure 1 and in supplementary Files S1, S2, S3, and S4) and I consider this work to be the heart of the manuscript, alongside showcasing how this information can be used to measure their abundances in biological samples.

We are very grateful to the Reviewer for the acknowledgement of our work on the fragmentation patterns of oxidized complex lipids. We hope that details provided in supplementary files of this manuscript will be of help for all the researchers aiming identification of oxidized lipids in variety of biological samples.

As I judge the topic of the manuscript to be highly relevant and the execution and interpretation of the LC-MS/MS to be highly skilled and detailed, I advocate for clarifying the method to increase applicability and thereby impact. My main suggestion is thus to provide the exact

methodology that the reader shall implement when aiming to apply it to their own samples. For large parts, the manuscript remains somewhat elusive as to what exactly the ‘developed methodology’ and the ‘holistic workflow’ entails. It is unclear what is meant by ‘biological context-specific’ profiling (as e. g. on p. 4) and it should be clarified why and at what stage samples should be pooled (p. 12). Figure 4A, for example, does not mention pooling of samples. Which extraction method and mobile phases do the authors recommend for different targets, especially in light of the variation with adduct type? Does this need to be optimized for each lipid class? Similarly, do the stepped CE values for the elevated energy HCD need to be adjusted?

To support the clarity of the presentation of our methodology, we included a new figure (Figure 4) in the revised version of the manuscript to provide detailed, step-wise description of the used workflow. Together with detailed figure legend, we hope that provided schematic workflow will be useful for the researchers aiming to use this new methodology in their own research.

Regarding the question about extraction/mobile phases – here we used relatively standard MTBE/methanol/water liquid-liquid extraction protocol and water/acetonitrile/isopropanol (supplemented with ammonium format and formic acid) mobile phase system with reverse phase chromatography. Thus, in our hands there is no need for specific optimization of extraction/LC conditions for epilipidomics experiments, at least for blood plasma samples, as well as for cell pellets and animal tissues such as mouse liver. Data on cell culture pellets and animal tissues are not shown here but we used these biological matrices in other experiments with the same types of analytical methodology.

Optimized here stepped collision energy values can be universally applied for different samples when data acquired at the same type of the MS instrument. Some adjustments might be necessary when method is transferred to other instrument types. For instance, currently we are using Exploris 240 instrument instead of QExactive Plus reported in the manuscript, and absolute or normalized collision energy values were changed according to the general recommendations provide by the vendor for between instrument method transfer.

Does the LPPTiger software generate the inclusion list for you?

Yes, new version of the LPPTiger (LPPTiger 2.0) supports inclusion list generation. Please see more detailed description of a new features included in the LPPTiger 2.0 in the section below.

A prior publication by the same group (reference 39) details the bioinformatic approach that “allows for the prediction of sample-specific oxidized lipidome” which is presumably what “in silico prediction” (e. g. p. 10) refers to. I laude the upload of (part of the) code to github but unfortunately and while the authors mention ‘high-throughput identification’ (p. 9), the data analysis and visualization strategy overall seems rather fragmented, using a number of software tools (LPPTiger, QualBrowser, Skyline, R, MetaboAnalyst etc.). This may exceed the scope of this publication, but I advocate for including as many steps as possible within GUI-based software for optimal reproducibility and dissemination. Perhaps a LPPTiger version 2 could include retention time mapping, KMD plots, etc. Along with that, do you recommend potential users to do a manual inspection as detailed on p. 8 for every new sample type and lipid class? I further wonder, whether the authors could comment on a possible implementation of this knowledge as on-line intelligent data acquisition? If not, what are the bottlenecks?

To increase the clarity of the workflow including data processing steps, a new figure (Figure 4) is now included in the revised version of the manuscript featuring not only analytical but also computational steps of our methodology. Briefly, LPPTiger 2.0 represents a key software as it supports *in silico* oxidation to predict sample specific epilipidome, generate inclusion lists

both for semi-targeted DDA and PRM data acquisition, and identification of oxidized lipids. Other software tools are used to support intermediate steps such as manual data inspection (QualBrowser), PRM based quantification (Skyline), and statistical data processing (MetaboAnalyst). All these tools are freely available, and widely accepted in metabolomics/lipidomics community. Thus, we prefer not to reinvent existing high quality solutions, but rather combine it with LPPtiger, which provides specific solutions for oxidized lipids, not available otherwise. In turn, following the Reviewer suggestion, we invested the time in providing new LPPtiger version (LPPtiger 2.0) supported by executable files and graphical user interface, which can be easily used by any researcher without bioinformatics background (see more details below).

On-line intelligent data acquisition solution would be of course a great tool to have! However, that would be a separate project which would require few years of work. However, the idea is very appealing, and we will consider it for the future steps. Possibly, oxidized lipids specific design of the acquisition methods similar to AcquireX workflow available now at certain ThermoScientific instruments would be a way to go.

Description of LPPtiger2 version and release of executable files for Windows platform:

Previous version of LPPtiger was significantly updated to support data processing within this study. However, majority of the new functions were only available with in-house customized LPPtiger source code through command line. To support the utility of the software for academic community, for the revised version of this manuscript we packed the source code into executable files version supported by graphical user interface, so that researchers without special bioinformatics expertise can freely use the software. LPPtiger 2.0 does not include retention time mapping and KMD plots, but these functionalities are included in our software development pipeline and will be available with the release of the next version.

LPPtiger 2.0 is now released and can be accessed via GitHub repository: <https://github.com/LMAI-TUD/lpptiger2>. The paragraph briefly describing new functionality of LPPtiger 2.0 is now included in the corresponding Methods section. Full user manual for LPPtiger 2.0 can be found at GitHub repository.

Lastly, I am not an expert on the biological example chosen and thus refrain from commenting in this regard. However, the number and extent of observed changes in the different groups is promising. A comparison to the direction and extent of changes observed in the corresponding non-oxidized species may add additional biological insights as well as showcase the need for measurements of oxidized complex lipids.

To address this question we performed a new set of experiments in which blood plasma lipid extracts prepared as before were used for identification of non-oxidized PC, CE, and TG lipids followed by quantitative lipidomics analysis in the same sample set of 150 lean, obese and obese with type 2 diabetes individuals. Obtained dataset was used for a correlation analysis against dataset featuring relative intensities of oxidized species. Details on LC-MS/MS methodologies used to acquire additional dataset are provided in Methods section of the revised manuscript together with data processing strategies (identification, quantification, statistical analysis). To illustrate our findings, a new Figure S5 is added to the revised version of the manuscript together with a short description of the results. Briefly, this correlation analysis clearly demonstrates that abundances of oxidized lipids both in lean, obese, and especially obese with type 2 samples do not correlated with corresponding non-modified lipids.

Similar, could the same conclusions have been drawn based on the oxylipin levels alone? Since the authors collected untargeted data, these comparisons may be possible without the

need for additional data collection. Are there any other quantified features that show significant differences between the groups, possibly even track with the oxidized lipids?

It has been repeatedly reported that in blood plasma 90 to 95% of oxylipins present in esterified form (esterified in complex lipids), with only a minor fraction present as free oxylipins. Thus, analysis of free oxylipins would require dedicated sample preparation which includes solid phase extraction to enrich very low abundant fraction of oxylipin followed by targeted LC-MS/MS (usually multiple reaction monitoring on triple quadrupole systems). Thus, we did not analyse here levels of free oxylipins but rather focused on their forms esterified in complex lipids such as PC, TG and CE. In the future, considering availability of both untargeted and targeted LC-MS/MS platforms in the same laboratory, complimentary analysis of esterified and free oxylipins can be performed.

Additional minor comments:

Presumably to circumvent the lack of available standards for oxidized complex lipids, the authors chose an 'in vitro oxidation' of PC lipid standards. While this certainly seems to be a viable approach, I am curious whether the authors could comment on how well this reflects naturally occurring oxidized lipids. Since we agree that one of the main challenges is chemical diversity of these lipids and that they may be partly enzymatically modified, I am wondering if this approach does not categorically exclude a subset of complex oxidized lipids that is not reproduced by the in vitro oxidation. In addition, am I assuming correctly that besides the MS analysis, we have no data to confirm the identity of these in vitro oxidized standards? In line with that, when the authors mention 'clean' spectra (p. 8), how can we be sure that they are produced by just one isomer?

Free radical driven lipid peroxidation, used here to generate *in vitro* oxidized lipid standards, is generally unspecific and results in a rich mixture of modified lipids, covering all types of modifications (from hydroperoxides to hydroxides, keto and epoxy derivatives). However, only some of those can occur via enzymatic reactions. Thus, *in vitro* generated oxidized standards represent comprehensive collection of everything what can chemically happen but not necessary happening in biological samples, where the process of lipid oxidation is more strictly constrained either by activity of specific enzymes or/and availability of free radicals. When we refer to "clean" spectra in this manuscript that means we can distinguish modification type (e.g. hydroperoxyl vs double hydroxy) and positional isomers (e.g. hydroperoxide at C15 position vs C5) isomers. Due to the lack of chemically defined standards for the majority of oxidized complex lipids, we utilized our and publicly available data on fragmentation patterns of oxylipins, many of which are commercially available. Modification type and position specific fragment ions of oxylipins are mirrored by the fragments formed by the corresponding acyl anion in complex oxidized lipids.

I may suggest a title that either focuses on the methodological approach or reflects the findings of applying this approach to the biological question. It is hereby of question whether the presented method should be referred to as 'holistic' when currently exclusively covering the three lipid classes of PC, CE, and TG lipids. Even if perhaps 'holistic' is referring to the depth of identification, I would suggest clarification of the terminology.

Title was changed to "Analytical and computational workflow for in-depth analysis of oxidized complex lipids in blood plasma of lean and obese individuals".

It may be advisable to refrain from using the term 'healthy' when referring to non-obese individuals (as on p. 12), not only because of the ongoing debate about obesity as a disease but also because of the potential confusion with diabetic status. From the methods and Table S4, it is unclear whether 'healthy' is describing anything beyond non-obese and non-diabetic.

Thank you noticing, it is a very valid suggestion. We substituted term “healthy” to “lean” in the revised version of the manuscript.

For Figure 1C, S3 and S4, the authors include NL as a measure of absolute intensity, for the sake of completion and comparability it would be good to also include this for oxPCs (in Figure 1 and File S2).

In revised version of the manuscript, we added NL values for all panels on Figures 1 and 2, as well as Files S3 and S4.

According to the caption, Figure 1A is supposed to show an MS3, however, in the figure it reads MS2 - this should be double-checked. As mentioned, a balance between absolute intensity and informative fragmentation needs to be struck, thus the reader would appreciate a measure of absolute intensity for the comparison of MS3 vs MS2 fragmentation.

We are not sure what reviewer meant here, as Figure 1A shows multistage MS3 spectra with upper panel representing 1st fragmentation event (MS2) and lower panel representing the 2nd fragmentation event (MS3). So we left it as it is. However, all NL values reflecting absolute signal intensities are now added to the figures.

In Figure 1E, the authors show the average abundances for 10 oxidized lipid species - is there a measure for how similar they behave, such as a standard deviation?

As requested, in the revised version of Figure 1E (and also 1E, as well as 3C and 3D) we added standard deviations. For the previous version of the Figures we did the calculations based on *in vitro* oxidized standard lipids. In revised version of the manuscript, we substituted these panels to the new one representing the distribution of different adducts and ISF values not only in *in vitro* oxidized standards but also blood plasma samples, to better reflect on the ionization preferences in different matrices.

The authors impressively show the impact of ISF, e. g. in Figure 3D. Perhaps a supplementary Figure could show a comparison in absolute intensity, so that one may gather the true impact. After identifying occurring ISF, did you alter the source settings to minimize it? If yes, which are the recommended settings?

To further illustrate the impact of ISF, in the revised version of the manuscript we included new supplementary Figure S2 with extracted ion chromatograms (in absolute intensities) for different adducts (NH₄⁺, Na⁺, H⁺) and corresponding ISF fragments for CE(18:2<OOH>)

For the source and ion optics we used settings optimized on our previous study [reference 38].

In Figure 3A, is smoothing applied? If trying to show chromatographic performance, it may be more appropriate to not, or at least mention whichever it is.

Yes, indeed Savitzky-Golay smoothing (5 points of window, polynomial order 2) was applied to XICs of oxCEs in the original manuscript. As the reviewer suggested, we replotted the graph with unprocessed XICs in the revised version of the manuscript.

In Figure 3B, the false positive IDs should be labeled as such.

Requested modifications were introduced into new version of Figure 3B.

The table in Figure 4B shows that about 1/3 of predicted lipids were identified. It would be interesting to know how many MS2s were triggered, i. e. the success rate of the identification. Is overlap in chromatography or quality of MS2s the main limiting factor? How close are we to

a holistic identification? How often was the initial identification correct compared to after manual inspection (p. 12)? Is there the potential to assign a score for future automation?

We consider identification of 1/3 of *in silico* predicted oxidized lipids in biological sample as a very high success rate attributed to the fact that *in silico* prediction is performed using sample specific lipidome information rather than generation of oxidized lipids list from randomly selected lipid species. Both chromatography separation (in case of multiple isomers, see answer to the Reviewer 1 on separation mono-oxidized arachidonate-PC isomers and Figure S1) and MS2 sensitivity (for very low abundant species) provides certain limitations, however the vast majority of initial identification were correct and manual inspection was mainly used to assign position specific fragments. Newly updated version of LPPtiger 2.0 which was made available with the revised version of the manuscript now supports major data processing steps, however manual inspection for position specific information and validation of identification results is still strongly advised (as for all other lipid identification tools) to ensure the quality of data annotation.

In the present manuscript, 17 PC, 17 TG, and 11 CE species were chosen as substrates for oxidation prediction. Is this the current limitation? I wonder if the authors could comment or speculate on what the theoretical or practical limit for this type of analysis is.

In principle, there are no limitation in the number of non-oxidized species to be used for *in silico* prediction. However, with increasing the number of non-modified lipids one would increase the number of the oxidized lipids for stDDA lists, so multiple injection would be required for complete identification runs. It is up to a user how much time and the sample he/she would like to invest in identification step.

Lastly, the manuscript and the clarity of its message would benefit from editing by an English speaker of native or native-like command and page and line numbers would make it easier for reviewers to comment precisely.

The revised version of the manuscript was proofread by the native English speaker.

Reviewer #3 (Remarks to the Author):

Here, Maria Fedorova and colleagues use mass spectrometry tools that they have developed in house to predict the likely oxygenated complex lipids that maybe present in plasma, and then they use this information to screen for changes in a large number of lipids in patient sample from healthy, obese and obese insulin resistant individuals. They find significant differences that could be very useful in terms of pushing forward what we know about lipid oxygenation in the circulation.

The study is conducted to a very high degree of technical accuracy, and the authors should be congratulated on this.

Thank you very much for this comment! We are happy you liked the work and appreciated all the technical details included in the manuscript.

I have three comments on the data itself.

1. The naming of the CE 18:1 epoxy... chemically, the convention is that the number coming after the colon includes the combined number of both rings and double bonds, not just double bonds. Thus, this should really be called 18:2 in line with the shorthand nomenclature of Liebisch et al.

Yes, notation of epoxy derivatives is rather confusing. Here we used updated rules for lipid shorthand notation (Liebisch et al., 2020, doi: [10.1194/jlr.S120001025](https://doi.org/10.1194/jlr.S120001025); see snapshot from the

corresponding table from the paper below). Till *sn*-position level, the number after the colon refers not to DB, but DBE, which includes DB, cyclic motifs (such as epoxy), and carbonyl C=O groups. When it comes to Structure Defined Level, the number you mentioned is just DB, and stating the modification (like epoxy, oxo, COOH) takes into account the corresponding part of total DBE.

Oxidative Modification	Species Level ^a	Molecular Species Level ^b	sn -Position Level ^c	Structure Defined Level
Hydroxylation	PC 36:4;O PC 34:1;O2	PC 16:0_20:4;O PC 16:0_18:1;O2	PC 16:0/20:4;O PC 16:0/18:1;O2	PC 16:0/20:4;OH PC 16:0/18:1;(OH)2
Epoxide	PC 34:2;O	PC 16:0_18:2;O	PC 16:0/18:2;O	PC 16:0/18:1;Ep
Hydroperoxide	PC 34:2;O2	PC 16:0_18:2;O2	PC 16:0/18:2;O2	PC 16:0/18:2;OOH
Peroxide	PC 34:2;O2	PC 16:0_18:2;O2	PC 16:0/18:2;O2	PC 16:0/18:1;OO
Aldehyde	PC 21:1;O	PC 16:0_5:1;O	PC 16:0/5:1;O	PC 16:0/5:0;oxo
Carboxylic acid	PC 25:1;O2	PC 16:0_9:1;O2	PC 16:0/9:1;O2	PC 16:0/9:0;COOH
Hydroxy-aldehyde	PC 26:3;O2	PC 18:1_8:2;O2	PC 18:1/8:2;O2	PC 18:1/8:1;OH;oxo

2. What form of plasma was used, EDTA, citrate or heparin? Was it taken in a way to minimise cell activation? If vacutainers were used, then activation of white cells and platelets can be a significant problem. How old was it? In storage for many years, oxidation can still happen, even at -80, so were all samples of a similar age in the cohort?

Updated text for Material and Method section is included in the revised version of the manuscript addressing Reviewer`s questions:

“EDTA plasma was collected for all samples in way that minimized cell activation, no vacutainers were used. Samples were always treated the same way, stored for less than 2 years, and provided for the analysis at a similar in “storage age.”

3. The heatmaps of plasma data are shown as log10 of normalised data, meaning that the reader can't make any easy comparisons between different lipids within the same samples, e.g. it isn't possible to tell which are more abundant than others. This is important since if the -OH forms of PE were primarily 12-HETE, or 5-HETE, then some speculation on their origin could be included. e.g. they may be considered to originate from platelets, or white cells and then perhaps they are present in extracellular vesicles instead of lipoproteins. On the other hand if there is an equal amount of all positional isomers, then this looks more like it could be driven by non-enzymatic oxidation. Hydrolysis and analysis of FA side chains for S/R ratios can also help but this may not be required if the relative abundance of positional isomers was known. This also applies for 9 and 13-HODE, in the case of LOXs and the CE isomers detected.

Thank you this comment, which prompted us to evaluate the data more closely. In the new supplementary Figure S4 we plotted absolute intensities of positional isomers for two HODE PC, two HETE PC, and HODE/HETE CE. For this, we used our PRM data and integrated peak areas corresponding to position specific fragment ions. Results turned out to be very interesting!

For modified PC lipids, indeed we see significant (multiple t-tests, parametric, unpaired) prevalence of 13-HODE over 9-HODE both for PC(16:0_18:2<OH>) and PC(18:0_18:2<OH>) in all three groups of samples (LND, OND, and OT2D) indicating involvement of 15LOX mediated enzymatic reactions (at least to some extent) in formation of these PC species. Specific enrichment of 13-HODE isomer was even more evident for CE(18:2<OOH>).

For HETE esterified PC, we observed different distribution of positional isomers for different PC species. Thus, for PC(16:0_20:4<OH>) species we quantified 3 groups of positional isomers represented by 15-HETE, 11/12-HETE and 7,8,9-HETE derivatives. Both 11/12-HETE and 7/8/9-HETE were significantly higher 15-HETE analogues in all studies conditions. Whereas for PC(18:0_20:4<OH>) we quantified two groups of positional isomers with 11/12-HETE containing species showing significantly higher intensities relative to 8/9-HETE isomers. This allows to propose the direct involvement of 12-LOX activity in their synthesis, in addition to non-enzymatic oxidation. Again, effect of non-enzymatic oxidation was clearly event for CE(20:4<OOH>), where 15-HETE, 12-HETE, and especially 9-HETE isomers were all abundantly present.

4. Moving forward from above, it would be very good if the data could be further interpreted biologically to ask (i) were enzymes involved or is this non-enzymatic, and (ii) where are the lipids, are they in lipoproteins, or cell derived EVs? I suspect that there maybe a different origin of the phospholipids (but perhaps the TG and CE are all in lipoproteins) due to the different FA composition that was seen.

Indeed, non-random distribution of positional isomers indicates involvement of specific enzymes in synthesis of detected oxidized lipids. Thus, for synthesis of HODE containing PC and especially CE (with higher levels in lean individuals) 15-LOX activity can be proposed. Whereas for HETE containing PC and CE 12-LOX as well as non-enzymatic oxidation are prevalent. Impact of non-enzymatic oxidation was clearly evident for CE(20:4<OOH>) with 9-HETE isomers being the most abundant one in OND and OT2D groups.

12-LOX derived lipids might originate from platelets but also adipose tissue derived extracellular vesicles. In fact, upregulation 12-LOX activity and elevated levels of 12-HETE were detected in visceral adipose tissue of diabetic subjects relative to obese individuals (PMID: 24955608). These, nicely correlated with results shown in this study for PC(18:0_20:4<OH>) for which 11/12-HETE PC isomer was significantly higher 8/9-HETE derivative and no other positional isomers were detected (Figure S4D). Importantly, this oxidized PC showed to be significantly enriched especially in OT2D group (Figure 5B).

On the other hand, as was mentioned by the Reviewer, neutral oxidized lipids originate most probably from lipoproteins. Interestingly, as already discussed in the manuscript, particular patterns can be observed here as well. Thus, oxCE enriched in lean individual indicate potential involvement of liver CYP450/sEH enzymes generating epoxy and diol derivatives. It also allows to propose that that this particulate oxCE might originate from VLDL lipoproteins.

Overall, despite multiple indications driven from this study on the source and specificity of modified lipids, we would like to avoid possible overinterpretations. In turn, these observations prompted us to initiate new experiments where we will apply the methodology reported in this manuscript to lipoprotein fractions isolated from human blood vs unfractionated plasma samples. This will allow to attribute the impact of each lipoprotein type to a particular modification type and its distribution as well as elucidate impact of different tissues to the circulating pool of oxidized lipids.

In our hands, lipid hydroperoxides are very unstable when trying to analyse them from actual biological samples. While we can extract them from buffer and measure them using LC/MS/MS very easily, we find that as soon as they are added to biological samples like disrupted cells, then they decompose rapidly (likely due to metal dependent redox cycling). Did the authors conduct controls to ensure that this was not a problem in their system and how did they minimise this? Were chelators added to prevent this?

We had worked with multiple biological matrices (blood plasma, tissue extracts, cell pellets) but did not face challenges with the detection of lipid hydroperoxides other than very prominent in source fragmentation. Here as an example where we demonstrate detection of CE(18:2<OOH>) standard spiked in to the blood plasma. Panel A demonstrates base peak chromatogram of blood plasma lipid extract prepared without CE(18:2<OOH>). Panel B shows base peak chromatogram of blood plasma spiked with CE(18:2<OOH>). Red trace illustrate extracted ion chromatogram for sodiated CE(18:2<OOH>) ion at m/z 703.5663. Panel C shows corresponding MS2 spectra.

As a possible troubleshooting, one could consider to monitor sodiated adducts instead of ammoniated, and to minimize in-source fragmentation by tuning ionization source setting as well as ion transfer optics.

REVIEWERS' COMMENTS

Reviewer #1 (Remarks to the Author):

The manuscript was carefully revised and new experimental data was included as requested. All of my questions were appropriately met and answered.

I do not have any further questions and recommend acceptance of the manuscript in its present form.

Reviewer #2 (Remarks to the Author):

All comments were sufficiently addressed and I have no further questions. A few direct comments below.

The added Figure 4 serves to clarify the workflow in a commendable fashion. I specifically would like to highlight the authors' distinction between optional and crucial (but freely available) software choices that should allow a wide variety of labs to implement the presented workflow.

It is a great addition to upload the software in a GUI format to GitHub and I would like to thank the authors for their work in this regard. As a brief note, in order to run LPPTiger on a Mac, I had to install rdkit, thus this may need to be added to the requirements. The example data worked in my hands. Since this is not the focus of the present manuscript and since there is a mechanism for feedback in place on GitHub, I will not further comment on the LPPTiger software itself.

I am happy to see the additional Figure S5 which convincingly shows the need for separate measurement of oxidized lipid species in the studied dataset.

Not including a comparison with free oxylipins is an acceptable limitation of the current study and I am looking forward to future comparisons as the authors suggested.

Reviewer #3 (Remarks to the Author):

The authors have answered my comments very well. I have one comment however. They talk about isomers of HETE being 12,15,11,7,8,9, however, I don't know about 7-HETE, and I don't think that's correct. Should this be 5-HETE?

Response to the Reviewers

We are thankful to all the reviewers for the constructive comments which helped us to improve the quality of the manuscript and we are happy to hear that Reviewers found the revised version of the manuscript ready for the publication.

Reviewer #1 (Remarks to the Author):

The manuscript was carefully revised and new experimental data was included as requested. All of my questions were appropriately met and answered.

I do not have any further questions and recommend acceptance of the manuscript in its present form.

We are glad that we could address all the comments stated by the Reviewer in the revised version of the manuscript.

Reviewer #2 (Remarks to the Author):

All comments were sufficiently addressed and I have no further questions. A few direct comments below. The added Figure 4 serves to clarify the workflow in a commendable fashion. I specifically would like to highlight the authors' distinction between optional and crucial (but freely available) software choices that should allow a wide variety of labs to implement the presented workflow.

It is a great addition to upload the software in a GUI format to GitHub and I would like to thank the authors for their work in this regard. As a brief note, in order to run LPPtiger on a Mac, I had to install rdkit, thus this may need to be added to the requirements. The example data worked in my hands. Since this is not the focus of the present manuscript and since there is a mechanism for feedback in place on GitHub, I will not further comment on the LPPtiger software itself.

I am happy to see the additional Figure S5 which convincingly shows the need for separate measurement of oxidized lipid species in the studied dataset.

Not including a comparison with free oxylipins is an acceptable limitation of the current study and I am looking forward to future comparisons as the authors suggested.

We are happy to hear that additional information and new version of LPPtiger software provided within the revised version of the manuscript was positively evaluated by the Reviewer.

Following Review suggestion, we additionally updated LPPtiger2 GitHub page and user guide file to include all dependencies of LPPtiger2 and more detailed description for source code installation steps.

Reviewer #3 (Remarks to the Author):

The authors have answered my comments very well. I have one comment however. They talk about isomers of HETE being 12,15,11,7,8,9, however, I don't know about 7-HETE, and I don't think that's correct. Should this be 5-HETE?

We are happy that our revised version of the manuscript addressed all comments of the Reviewer.

Regarding 7-HETE isomers:

We agree that 7-HETE isomer is rather unusual. With this in mind, we carefully reviewed our dataset. However, we still believe that we detect certain amount of this isomeric product in blood plasma lipid extracts, based on the following observations:

- We are convinced that reported isomer is not 5-HETE derivative of PC. PC(16:0_20:4<OH{5}>) was quantified as a separate peak with the latest eluting time among PC(16:0_20:4<OH>) isomers (Δ RT 0.6 min between 7, or 8, or 9-HETE PC and 5-HETE PC). However, PC(16:0_20:4<OH{5}>) was filtered out during data post-processing as it was quantified only 1, 5, and 4 samples out of 50 for each group (LND, OND, and OT2D, respectively).
- Identification of 7-HETE derivatives of PC was based on the presence of fragment ion at m/z 141.0557 (C₇H₉O₃⁻). The formation of C₇H₉O₃⁻ fragment ion is in agreement with fragmentation rules defined for various regioisomers described in Supplementary Data 1. This fragment ion was not detected in any other isomeric species including regioisomers of FA(20:4<OH>) with OH in position {5}, {8}, {9}, {11}, {12}, {15}, and FA(20:3<ep>) with ep in positions {5-6}, {8-9}, {11-12}, {14-15}.
- In addition to PC(16:0_20:4<OH{7}>), we detected the presence of PC(16:0_20:4<OOH{7}>) and PC(18:0_20:4<OOH{7}>) in blood plasma of OND and OT2D individuals (see p. 35, 53, 80, and 102 of Supplementary Data 6 for MS/MS spectra).

Overall, such “unconventional” oxygenation at position 7 is an example of a reaction under kinetic control, which means that the product forms faster but it is less stable thermodynamically than regioisomers at positions 5 or 9 (thermodynamic products) [Yin et al., 2011; PMID: 21861450]. Such kinetic products are hard to detect, because lipid peroxidation occurs usually under thermodynamic control. But in the presence of good H-donors (many natural antioxidants) that can “trap” fast-forming products, kinetic products (such as 7-HETE/7-HpETE) can be detected. This brings us to an interesting conclusion. The fact that we were able to see esterified 7-HETE and 7-HpETE (although their quantities are presumably low compared to thermodynamic regioisomers) marks activity of the antioxidant defence system upon lipid peroxidation. However, this assumption requires further validation.